# Inconsistent shifts in warming and temperature variability are linked to reduced avian fitness

Conor C. Taff [1,3] ✉ & J. Ryan. Shipley [2,3]

As the climate has warmed, many birds have advanced their breeding timing. However, as climate change also changes temperature distributions, breeding earlier might increase nestling exposure to either extreme heat or cold. Here, we combine >300,000 breeding records from 24 North American birds with historical temperature data to understand how exposure to extreme temperatures has changed. Average spring temperature increased since 1950 but change in timing of extremes was inconsistent in direction and magnitude; thus, populations could not track both average and extreme temperatures. Relative fitness was reduced following heatwaves and cold snaps in 11 and 16 of 24 species, respectively. Latitudinal variation in sensitivity in three widespread species suggests that vulnerability to extremes at range limits may contribute to range shifts. Our results add to evidence demonstrating that understanding individual sensitivity and its links to population level processes is critical for predicting vulnerability to changing climates.

Human induced climate change has resulted in an accelerating increase in the average temperatures that animals experience around the globe[1]. A rich literature now documents the consequences that span from the scale of individuals to entire populations and species[2]. The most universal observed responses to warming are reductions in body size[3–5], shifts in species ranges[2], and changes in the timing of annual events (i.e., phenological shifts)[6–8], which have the potential to impact fitness via increased interspecific competition[9] or phenological mismatches between predators and prey[10–13]. However, mean temperatures alone fail to capture the complexity of climate change, as the rate and magnitude of warming often differs between day and night, as well as seasonally. Another component of climate change is increasing temperature variability, which is predicted to have greater impacts on species survival than increases in mean temperature[14]. As the pace, magnitude, and variability of climate change determines the risk of exposure to conditions outside those historically experienced, there is a critical need to understand how and why the magnitude of these consequences differs between species to predict and mitigate the effects of climate change for wild animals[15].

Recently, Rosenberg et al.[16] estimated that the total population of North American birds has declined by almost 3 billion since 1970. While these declines arise from a combination of several factors (e.g., introduced predators[17]; noise and light pollution[18]; land use change[19]; disease[20]), climate change is generally considered to be one of the primary threats and drivers[21,22]. However, the declines in overall abundance are far from uniform, with some groups, such as aerial insectivores, showing steep declines while others, such as many wetland birds, have increased in numbers[16]. These differences suggest that life history and habitat requirements may play an important role in determining the sensitivity of different species to climate change.

One of the major goals of many climate studies has been to assess whether wild birds are shifting their ranges or advancing breeding phenology fast enough to track changes in mean temperatures[23–25], with several recent studies suggesting that observed changes are generally inadequate to keep pace[26–28]. However, it is also increasingly apparent that changes in mean temperatures are insufficient to understand the response to climate change, as the mean only provides

[1]Department of Ecology & Evolutionary Biology and Lab of Ornithology, Cornell University and Biology Department, Colby College, Waterville, ME 04901, USA. [2]Swiss Federal Institute for Forest, Snow and Landscape Research WSL, Zürcherstrasse 111, 8903 Birmensdorf, Switzerland. [3]These authors contributed equally: Conor C. Taff, J. Ryan. Shipley. ✉e-mail: cct663@gmail.com

a coarse approximation of changes in minimum (nighttime) and maximum (daytime) temperatures, which can change independently in pace, magnitude, and variability. Different organismal traits such as timing of reproduction or when a species is active likely shape the response because animals may need to balance changing temperatures simultaneously with other abiotic gradients, such as day length, UV radiation, or water availability[29,30]. Even when only considering mean temperatures, perfectly tracking average daily temperatures may result in a more variable thermal environment when breeding activities are occurring earlier in the year[31] or when climate warming is associated with increased climate variability[32–34]. When predictable conditions are linked to survival, increases in either the magnitude or the frequency of conditions outside historical norms can result in reduced fitness and declines at the population or species level.

The relative importance of different abiotic gradients for each species and their link to fitness, coupled with the pace of plasticity or evolutionary change, likely dictates which species persist and which will perish[22]. The strength of this relationship is determined by how efficiently organisms transfer heat and energy in different environmental conditions and how these traits influence individual performance as well as the ability to survive, develop, grow, and reproduce[35]. For example, several studies demonstrate that life history traits predict different aspects of the response to climate change[19,36,37]. Among birds, foraging mode is one key life history characteristic that might influence the relative challenge posed by average temperature versus variation in temperature. The vast majority of birds are reliant on insects to provide energy and nutrients to their rapidly developing offspring. For example, obligate aerial insectivores depend on flying insects, even modest decreases in daytime temperatures can reduce insect activity and thereby overall foraging success[38] (hereafter "cold snaps"). If these reductions in insect activity occur during critical periods when nestlings are most vulnerable, they can trigger mass mortality events[31]. Accordingly, species that do not rely on insect activity while foraging are likely to be less sensitive to cold snaps. In contrast, heatwaves would be expected to impact a broader range of species as they may approach physiological upper temperature limits[39,40]. A more thorough understanding of relative sensitivity to average temperature, cold snaps, and heatwaves during breeding attempts requires data that spans multiple species across a wide range of conditions paired with detailed records of reproductive performance.

The most detailed studies of climate change on populations tend to focus on single species with a long time series of detailed breeding data[3,31,32,41]. These studies provide evidence for the mechanisms by which climate change influences populations, but they typically do not address variation in responses between species or different populations. Another approach is to use large scale observations to model spatial, temporal, and cross species variation in population phenology[7,8], abundance[16], body size[4,42], or ranges[43,44] in response to climate change. However, these studies typically cannot link the patterns observed at coarse scales to specific processes and mechanisms that occur at the level of individual animals[25]. Linking individual fitness to population level processes with large scale and spatially variable outcomes is needed to accurately predict variation in sensitivity to climate change across species.

In this study, we used a database of community-contributed observations on the timing of breeding and reproductive success in wild North American birds compiled by Project MartinWatch, by the NestWatch program at the Cornell Lab of Ornithology, and by Project Nestwatch from Birds Canada between 1995 and 2020. We identified >300,000 breeding records from 24 common species with nesting records spanning most of the United States and Canada. By documenting the reproductive performance of individuals, this database allows for detailed exploration of spatial, temporal, and interspecific variation in temperature exposure and vulnerability at near continental scales.

We first asked how the change in timing of cold snaps and heatwaves compares with changes in overall average spring temperature across the range of locations from which breeding data were available. If these abiotic gradients have advanced at different paces or in different directions, then populations will be unable to track the same breeding conditions regardless of phenological advancement. Next, we determined the temperatures experienced during every nesting attempt and asked whether each species showed evidence of performance declines during cold snaps or heatwaves during the breeding attempt (i.e., whether these events qualify as extreme climatic events 'ECEs' with biological consequences[45,46]). We predicted that performance declines would be most pronounced for cold snaps experienced by aerial insectivores because of the direct impact on flying insect availability. In contrast, we predicted that all species in the dataset would be vulnerable to heatwaves.

Despite an increase in average spring temperature, we found that the date of the latest cold snap and earliest heatwave has not changed consistently over the last 70 years. Nesting attempts that occurred during a cold snap or heatwave were associated with reduced fitness in 11 and 16 out of 24 species respectively. Our results demonstrate that exposure and sensitivity to cold snaps and heatwaves during the breeding season may be an important component of vulnerability to climate change.

## Results
### Spring temperature and cold snap or heatwave timing
Across the spatial range that we studied, the average spring temperature anomaly from 1995 to 2020 was universally positive, although there was spatial variation in the magnitude of this increase in average spring temperature (Fig. 1A). In contrast, change in the date of the latest 3-day cold snap or the earliest 3-day heatwave was inconsistent in both sign and magnitude (Fig. 1B, C).

In a spatial GAM averaging across the entire range studied, spring temperature anomaly was consistently positive from 1995 to 2020 and the last cold snap tended to occur 3 to 5 days earlier than the reference period regardless of the cold snap threshold used (Fig. 2A, B). However, the first heatwave was not consistently different from the reference period regardless of the temperature threshold used (Fig. 2C).

When considering average anomalies over the last 25 years, there was no evidence that a larger overall spring temperature anomaly was associated with any consistent difference in the change in cold snap timing (Fig. 3A) or heatwave timing (Fig. 3B). When comparing the temperature anomaly in each individual year at the level of hexagonal grids to the cold snap or heatwave anomaly in each individual grid-year, there was an association such that grid-years with higher spring temperature anomalies tended to have both earlier last cold snaps and earlier heatwaves (Fig. S5), but there was wide variation for individual grid-years.

### Impact of cold snaps and heatwaves on reproductive success
Because of differences in range and breeding timing, the species in our dataset varied widely in their exposure to temperature during breeding (Fig. 4A). During incubation, point estimates from GAMs that controlled for date, year, and location, indicated that a two standard deviation cold snap reduced relative fitness in 8 out of 24 species (Fig. 4B). The species that were sensitive to cold snaps in this period included purple martins (*Progne subis*), tree swallows (*Tachycineta bicolor*), eastern bluebirds (*Sialia sialis*), western bluebirds (*Sialia mexicana*), mountain bluebirds (*Sialia currocoides*), prothonotary warblers (*Protonotaria citrea*), Carolina chickadee (*Poecile carolinensis*), and mountain chickadees (*Poecile gambeli*). Relative fitness estimates ranged from 0.64 to 0.93 for these species; all values and confidence intervals for point estimates are included in Table S2. Only two of these same species also showed evidence of reduced fitness during a two SD incubation heatwave (purple martin and eastern bluebird; Fig. 4B, Table S2). No species showed

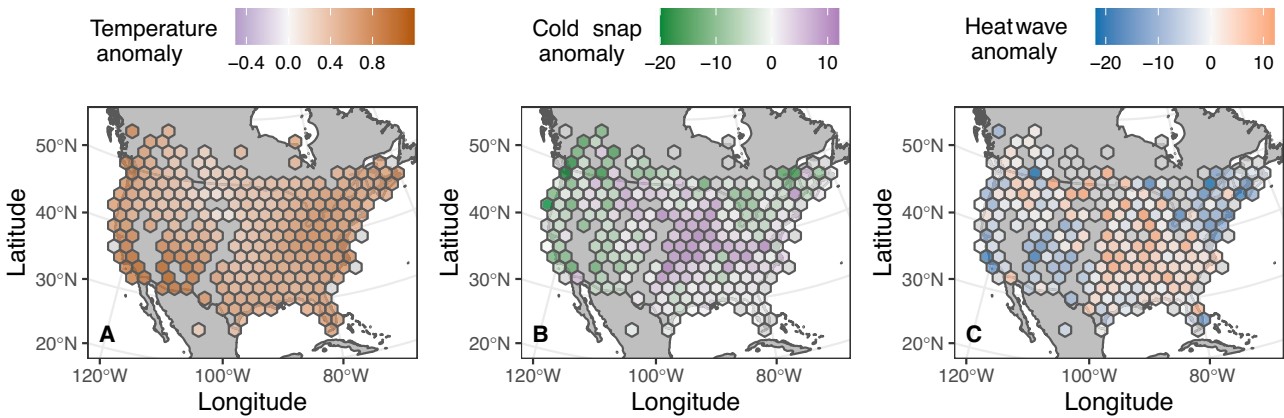

**Fig. 1 | Spring temperature anomaly (A), cold snap anomaly (B), and heatwave anomaly (C) for the North American spatial extent studied over the years 1995–2020.** Spring temperature anomaly is defined as the average of April, May, and June using data from the Berkeley Earth Project measured in degrees C compared to the baseline period of 1950–1980. Cold snap anomaly is the difference in average timing of the latest three-day period in which temperatures did not rise above 17.4 C from 1995 to 2020 compared to the average date of the latest cold snap from 1950 to 1980, measured in days. Heatwave anomaly is the difference in timing of the earliest three-day period in which temperatures always rose above 32.1 C from 1995 to 2020 compared to the average date of the earliest heatwave from 1950–1980, measured in days. Gray cells indicate regions with missing data.

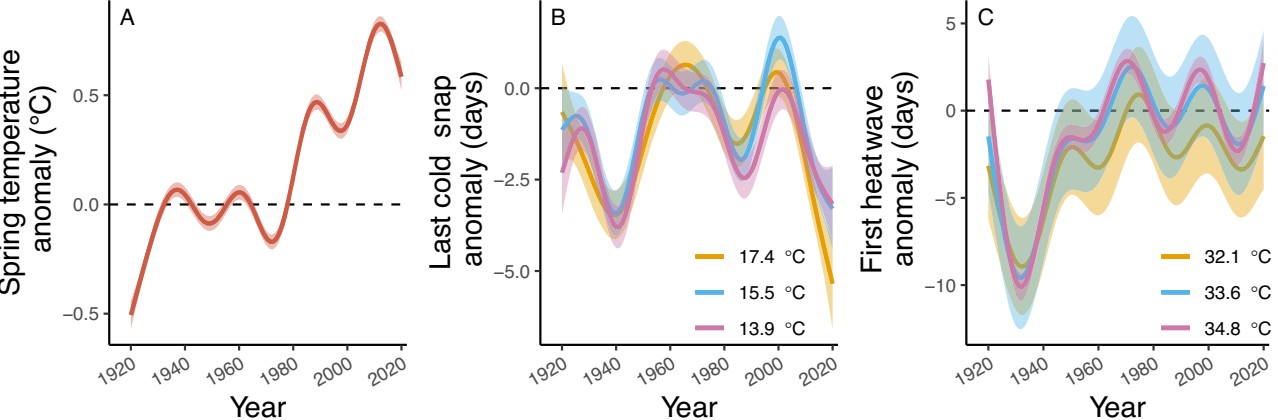

**Fig. 2 | The historical change across the North American spatial extent studied in average spring temperature anomaly (A), timing of the last cold snap (B), or timing of the first heatwave (C) over the last 100 years.** Lines and 95% confidence intervals are from spatial GAMs that account for latitude and longitude of each grid cell. In all three panels, solid lines represent the point estimate values for the model and shaded regions represent the 95% confidence intervals for those estimates. For the cold snap and heatwave panels, three different threshold values are shown to illustrate increasingly more severe cold snaps or heatwaves.

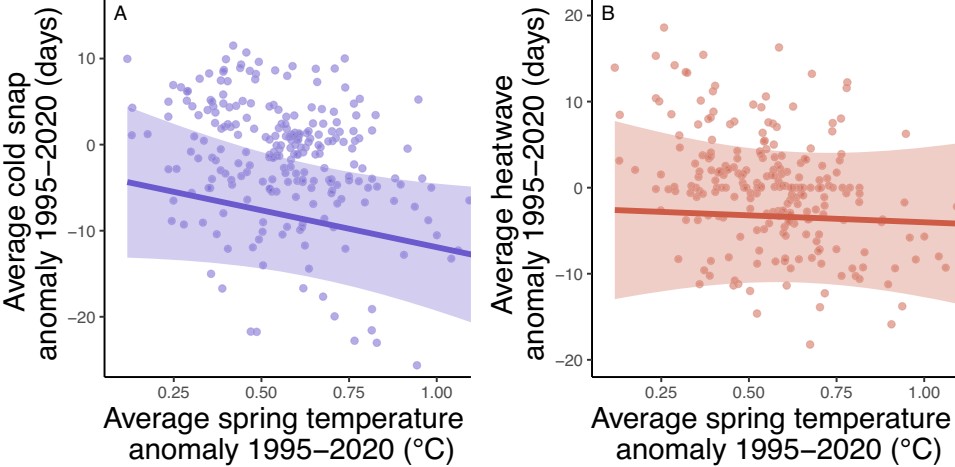

**Fig. 3 | Relationship between average spring temperature anomaly for the North American spatial extent studied for each grid cell from 1995 to 2020 and (A) average cold snap anomaly or (B) average heatwave anomaly over the** same period. Points are the average values for each hexagon grid cell. Lines and 95% confidence intervals are derived from GAMs that include a spatial smooth for latitude and longitude to account for spatial autocorrelation.

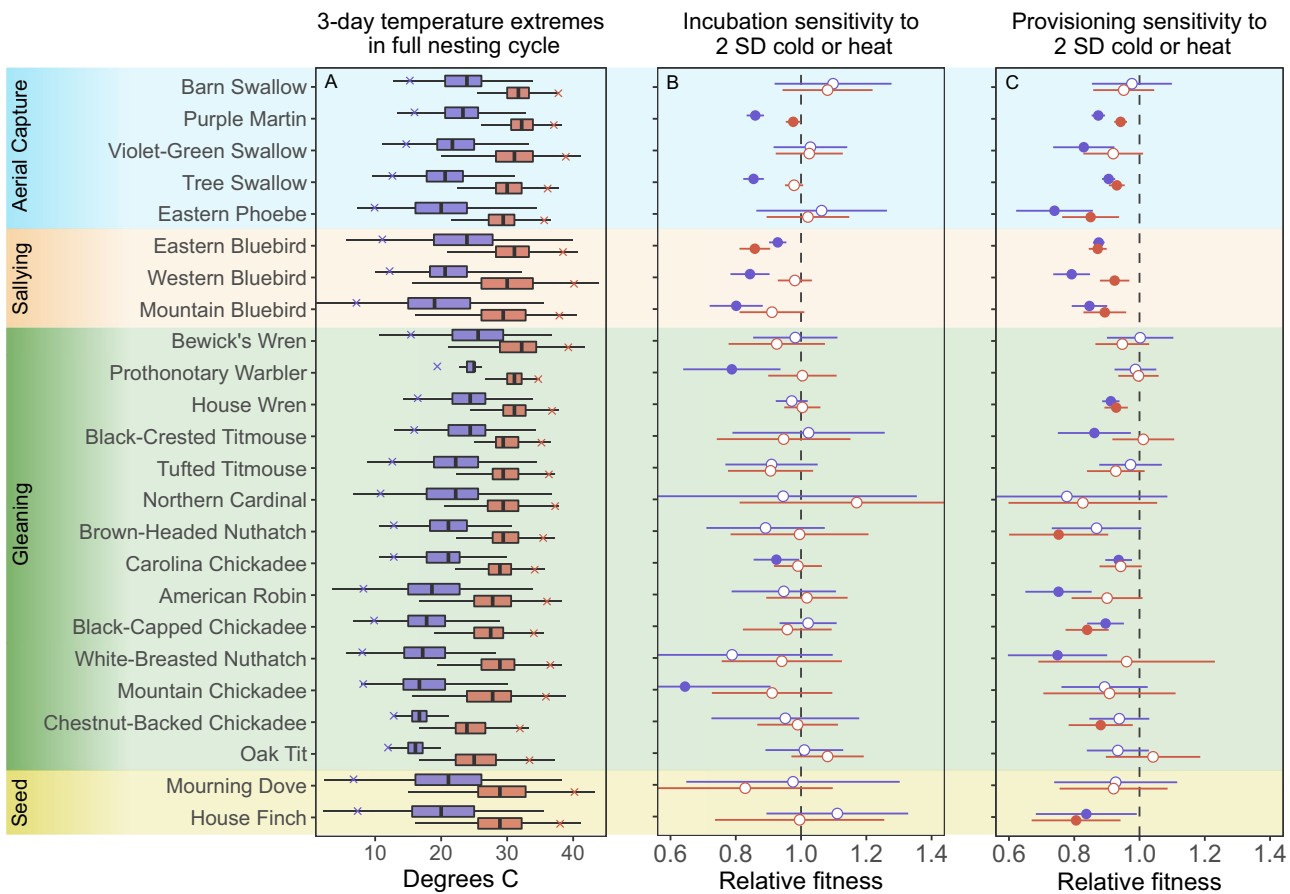

**Fig. 4 | Exposure and sensitivity to cold snaps and heatwaves across 24 species.**
**A** Distribution of the coldest and hottest 3-day high temperatures experienced during the full nesting cycle for all species sorted by the coldest average temperature within each foraging style. Boxplots indicate median, IQR, and 1.5 times IQR for the coldest (blue) and hottest (red) 3-day period. Blue and red x marks indicate points that are 2 SD below (cold) or above (hot) the mean exposure. **B**, **C** Predicted relative fitness for breeding attempts that experienced a 2 SD cold snap (blue) or 2 SD heatwave (red) during incubation (**B**) or provisioning (**C**). Point estimates are derived from GAMs that control for spatial patterns, year, and date. Circles show point estimate, lines show 95% CI; open circles indicate no difference from mean reproductive success while closed circles indicate a significant effect of temperature on relative reproductive success. Color bands show the primary foraging strategy that each species uses to provision nestlings. The number of biological replicates (nests) used to derive the values for each species during incubation and provisioning is provided in Table S1.

evidence of significantly increased fitness from either cold snap or heatwave exposure during incubation.

During the nestling provisioning period, 75% of the species (6 of the 8) that were sensitive to cold snaps in incubation also showed reduced relative fitness from two SD cold snaps occurring after hatching (prothonotary warbler and mountain chickadee were no longer susceptible; Fig. 4C). In addition to these six species, eastern phoebes (*Sayornis phoebe*), violet-green swallows (*Tachycineta thalassina*), and six other species had reduced fitness when a two SD cold snap occurred during the provisioning period (Fig. 4C). Overall, the effect sizes for reduction in relative fitness during provisioning were comparable to those seen during incubation (range of estimates for susceptible species was 0.74 to 0.94; Table S2). Nine of the 14 species susceptible to cold snaps were also susceptible to heatwaves during provisioning (Fig. 4C). Two additional species (brown-headed nuthatch: *Sitta pusilla* and chestnut-backed chickadee: *Poecile rufescens*) were susceptible to heatwaves but not cold snaps during provisioning (Fig. 4C). Similar to the incubation phase, no species showed evidence of significantly increased fitness from either cold snap or heatwave exposure during the nestling phase.

**Latitudinal variation in temperature exposure and susceptibility**
We examined latitudinal trends in susceptibility to cold snaps and heatwaves for eastern bluebirds, purple martins, and tree swallows.

In all three species, breeding date was later farther north, but breeding attempts from northern areas still experienced lower 3-day coldest and 3-day hottest temperatures on average (Fig. 5A, B, Figs. S6A, B and S7A, B). While the overall patterns of susceptibility to cold snaps and heatwaves were largely similar across these species (Fig. 4B, C), the latitudinal patterns differed somewhat for each species. During both incubation and provisioning, eastern bluebirds had reduced fitness from cold snaps only near the northern edge of the range (Fig. 5C, D; full details on point estimates in Table S3). In contrast, relative fitness was reduced from heatwaves over a wider, but somewhat inconsistent latitudinal extent (Fig. 5C, D).

Purple martin susceptibility to heatwaves was only apparent near the southern edge of the range and was more pronounced in provisioning than during the incubation period (Fig. S6C, D). Unlike eastern bluebirds, purple martins had reduced relative fitness from cold snaps during incubation and provisioning at nearly every latitude band. Only the southernmost two bands during provisioning showed no impact on fitness associated with cold snaps (Fig. S6C, D; Table S4).

Like purple martins, tree swallows had consistently reduced relative fitness when cold snaps occurred during incubation (Fig. S7C). Despite an aggregate effect of heatwaves during incubation (Fig. 4C), there was no signal for heatwave effects during incubation or provisioning in any individual latitude band (Fig. S7C, D). During provisioning, tree swallows only showed a sensitivity to cold snaps at the

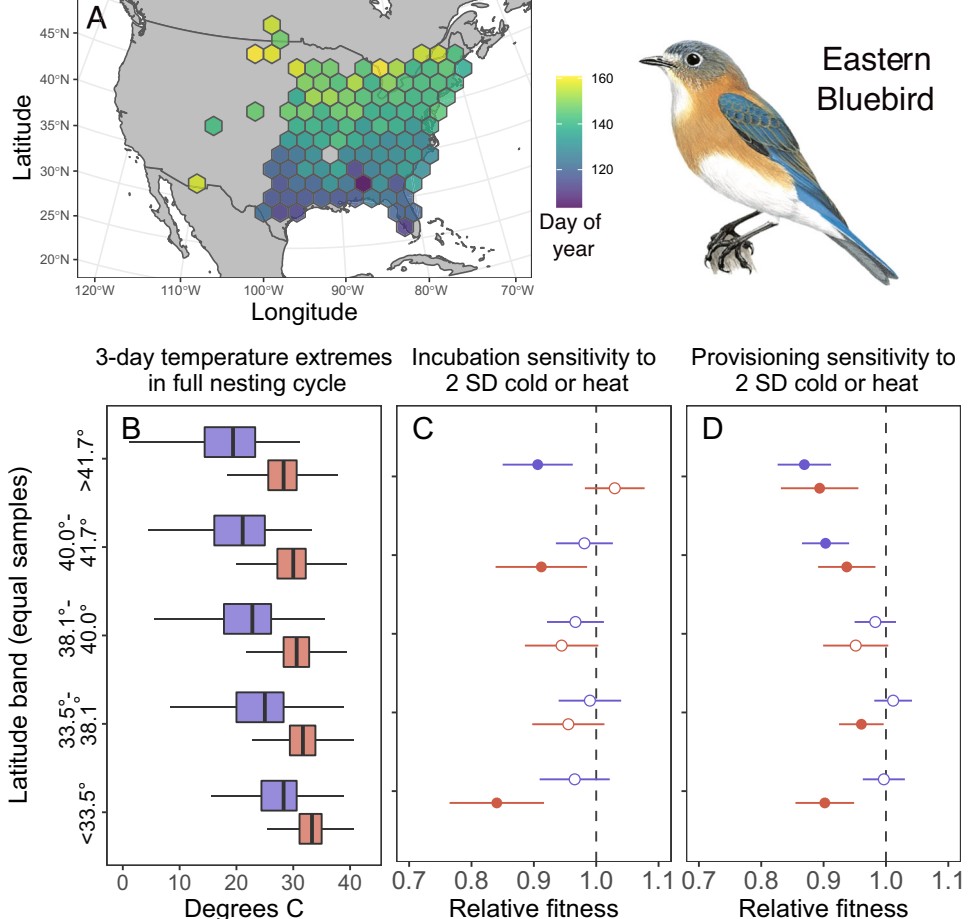

**Fig. 5 | Breeding timing, temperature exposure, and extreme temperature sensitivity in eastern bluebirds. A** Average date of first egg laying across the records included. **B** Distribution of the coldest and hottest 3-day high temperatures in the full nesting cycles in five latitude bands with an equal number of samples per band. The top of the panel is the northernmost band and bottom is the southernmost. Boxplots indicate median, IQR, and 1.5 times IQR for the coldest (blue) and hottest (red) 3-day period. **C, D** Predicted relative fitness for breeding attempts that experienced a 2 SD cold snap (blue) or 2 SD heatwave (red) during incubation (**C**) or provisioning (**D**) for each latitude band. Point estimates are derived from GAMs that control for spatial patterns, year, and date. Circles show point estimate, lines show 95 percent CI; open circles indicate no difference from mean reproductive success while closed circles indicate a significant effect of temperature on relative reproductive success. In panels **B** and **C**, the number of nests used to derive estimates from northernmost to southernmost latitude bands was 9254, 9897, 10589, 11363, and 12321, respectively. In panel **D**, the number of nests included was 8035, 8657, 9244, 9990, and 10662, respectively. Illustration from Handbook of the Birds of the World by Ian Willis, copyright Lynx Edicions.

northern range limit and no clear susceptibility to heatwaves (Fig. S7D; Table S5).

## Discussion

Using community scientist collected breeding records, we show that exposure to three-day cold snaps during a breeding attempt is associated with reduced relative fitness in 16 of 24 common species and that exposure to three-day heatwaves is associated with reduced fitness in 11 of 24 species. In three widespread species, sensitivity to cold snaps and heatwaves was spatially heterogeneous, indicating a mechanism by which climate change might generate population trends that differ across a species range. Historical temperature records for the same area showed that the date of the latest cold snap is only slightly earlier, and the date of the earliest heatwave has not changed consistently over the past 70 years despite warmer springs overall. Taken together, our results demonstrate that the likelihood of encountering cold snaps or heatwaves during breeding might differ as a consequence of climate change and that exposure to these events often results in reduced reproductive success for many common and widespread species. Moreover, in many cases it is impossible to track both average temperature and temperature variability through

phenological shifts in the timing of breeding. As a consequence, species that are most sensitive to extreme temperature events may be less able to adapt to changes in average temperature. Within species, variation in sensitivity at northern and southern range limits might contribute to range shifts and local population declines. Understanding the mechanisms that result in different sensitivity to temperature variability within and between species should help accurately predict which populations are most vulnerable to climate change.

Across the range of breeding records that we studied, average spring temperatures have increased over the last 70 years, but the average timing of latest cold snaps and earliest heatwaves during the breeding season has not changed consistently in sign or magnitude. As a consequence, the historical association between average temperature and temperature variability differs from the association observed over the last 25 years. Thus, cues that wild birds use to time seasonal events may not contain the same information about average temperature and temperature variability that they did historically. Changes in breeding phenology have been identified as one of the 'universal responses' to climate change[47,48]. In some cases, these shifts can minimize the effects of phenological mismatch with food resources[6,49-51], but in many cases the degree of change in

breeding timing seems inadequate to keep pace with average temperatures[23,26,27]. These inadequate shifts may be attributable to a lack of ability for rapid evolutionary change or incomplete phenotypic plasticity[52,53]. However, average breeding temperature and food availability are not the only gradients that could be disrupted by climate change[29]. For species that are especially susceptible to cold snaps or heatwaves during breeding, changes in the timing of breeding events to track average temperature might result in increased exposure to temperature variability[31,34]. Our results highlight the consequences of temperature variation and suggest that in some cases incomplete plasticity and a degree of phenological mismatch may represent an adaptive response that balances competing selection pressures[54].

We predicted that the consequences of cold snaps might be especially severe during the nestling provisioning period and for species that rely on insect activity for foraging. A recent meta-analysis demonstrated that environmental insect food abundance is a strong predictor of nestling body condition and fledging success, especially in species in which insects make up a large component of the diet[55]. Daily total flying insect biomass and emergence rates are strongly influenced by temperature[38,56], so cold snaps can have immediate and direct effects on food availability. These impacts on food availability could compound the increased thermoregulatory challenges and associated increase in energy use during cold snaps. We did find that more species were sensitive to cold snaps during nestling provisioning, although several species were sensitive in both stages (6 species in both stages, 2 species in incubation only, 8 species in provisioning only). The 14 species that were sensitive to cold during provisioning included four of the five obligate aerial insectivores (purple martin, tree swallow, violet-green-swallow, and eastern phoebe) and all three species that rely extensively on insect movement for sally capture foraging (eastern, western, and mountain bluebird). However, white-breasted nuthatches (*Sitta carolinensis*) and American robins (*Turdus migratorius*) also showed declines despite a less obvious link to insect availability and the other aerial insectivore did not exhibit sensitivity (barn swallow, *Hirundo rustica*). Thus, our results suggest the possibility of foraging mode as a mechanism generating susceptibility to cold snaps during provisioning, but more data for a broader distribution of phylogenetically independent species with different foraging styles would be needed to evaluate this prediction convincingly.

In contrast to cold snaps, we predicted that the primary challenge from heatwaves would be their impact on thermoregulation, rather than indirect effects of food availability. For example, several studies demonstrate that heat dissipation rate may constrain reproductive investment[57,58] and hot conditions increase cooling costs[59]. Therefore, we expected to find more widespread evidence for heatwave impacts during both incubation and provisioning. While we did find fairly widespread effects of heatwaves, fewer species were sensitive to heatwaves than to cold snaps (11 versus 18 species across both stages). For nine of these species the sensitivity was only observed in one breeding stage. Moreover, the effect sizes for reduction in relative fitness from heatwaves were generally smaller than those observed for cold snaps. While natural heatwaves during breeding can clearly drive reduced fitness in some cases[60,61], over the range and set of species we studied, cold snaps seemed to generate more consistent and stronger reductions in breeding success even though spring breeding conditions are warmer overall than they were historically. One reason for these smaller effects might be that the species in our study were able to mitigate the risks of hyperthermia by changing behaviors or by choosing nesting sites that were cooler than the overall temperature recorded from nearby weather stations. It is also possible that a different characterization of heatwaves, such as one using average rather than daily high temperature or a longer time period than three days, might have detected more vulnerability.

In one sense, it is surprising that we did not find universal evidence for cold snap and heatwave consequences, because ultimately it is clear that a sufficiently severe temperature extreme would reduce fitness. Several nonmutually exclusive factors likely explain the lack of evidence for sensitivity that we observed in some species. First, species ranges are shaped, in part, by direct and indirect effects of temperature[62]. Because we took the approach of generating point estimates that were 2 SD from the mean of the observed breeding records for each species, the absolute value of temperature challenges that we modeled differed across species. While our approach explores the realized fitness declines from actual temperature exposure, manipulations or extensions beyond these temperature extremes would eventually uncover sensitivities in all species. Many experimental studies have demonstrated physiological or fitness costs from cooling[63,64] or heating[65,66] nest boxes during breeding, but these often employ more sustained or extreme temperatures than the natural variation we modeled. On a related note, some species with relatively restricted ranges limited by temperature exposure might be highly sensitive to temperature extremes, but rarely experience those conditions during breeding, preventing us from detecting effects of temperature challenges. Pigot et al. 2010 argued that widespread species range limits are more likely to be defined by temperature, so it may be more common to detect sensitivities to temperature extremes in those same widespread species[62]. Finally, the community science datasets that we used yielded highly variable sample sizes and do not necessarily include representative observations across the range of most species. For widespread and easily observed species it is likely easier to detect effects both because of larger sample sizes and because of more dispersed sampling across the extent of each species range. Samples near the range limits might be especially important for detecting sensitivity to temperature extremes and are not equally available for all species in our dataset (e.g., bluebird trails and nest boxes make monitoring especially easy and widespread for these species).

The results from our more detailed analysis of records from eastern bluebirds, purple martins, and tree swallows do indeed suggest that within-species spatial dynamics in sensitivity to temperature variability may be highly relevant. Eastern bluebirds were vulnerable to heatwaves over most of their range especially during provisioning but were only sensitive to cold snaps near the northern range limit. Purple martins were vulnerable to cold snaps across their range during incubation and in the northern half of the range during provisioning but were only sensitive to heatwaves near their southern range limit. Tree swallows had no clear sensitivity to heatwaves within narrow latitude bands; they were sensitive to cold snaps in incubation across most of their range, but only sensitive to cold snaps during provisioning near their northern range limit. We did not quantify overall abundance changes in our study, but it is interesting to note that since 2007 purple martins have declined near their southern range limit and tree swallows have declined most precipitously near their northern range limit while expanding their range southward[67]. These patterns qualitatively match with the range limits near which we found differences in sensitivity for each species. In contrast, eastern bluebirds have declined west of the Appalachian Mountains and increased on the Atlantic seaboard over the same period, so it is less clear how the patterns might relate to the sensitivity differences we found[67]. Nevertheless, the fact that all three species show at least some evidence of increased sensitivity to temperature near a range limit suggests that exposure to increased temperature variability might be an important contributor to range shifts with climate change. However, despite the fact that all three species have similar body sizes, breeding behavior, and distributions, with two being fairly close relatives that are very similar ecologically, there was no common latitudinal pattern. Presumably these idiosyncratic responses are driven by subtle differences, such as exposure to multiple additive stressors, life history details, or habitat requirements. Thus, predicting differential sensitivity to temperature variability will likely require a detailed understanding of the ecology for each species.

One important limitation on the conclusions of our study is that our data only allowed us to model overall fledging success, but it is likely that temperature variability also generates sub-lethal effects that could have long term consequences on population demographics. Early developmental conditions, including temperature, are well known to have a wide variety of long lasting effects on wild birds, such as changes in body size and morphology, physiology, immunology, and survival to recruitment as a breeding adult[68]. Indeed, long term declines in body size is another widespread consequence of global warming[5]. While changes in body size may sometimes result from rapid evolutionary responses[4], many of the observed changes in body size could also be explained by changes in developmental temperature[69]. For example, in tree swallows fledgling body size is positively correlated with average developmental temperature and predicts both adult body size and likelihood of recruiting[69]. Thus, cold snaps or heatwaves that are not severe enough to reduce fledging success could still have important consequences on the demographics of bird populations through their long term effects on nestlings. Our results should therefore be considered as a minimum measure of the fitness costs from experiencing extreme temperature conditions.

A great deal of research has focused on the effects of climate change on wild bird populations over the past several decades. While the large-scale patterns of climate change on ranges, phenology, and morphology have been described for many species, studies that focus on these patterns are often unable to characterize the organismal and ecological processes that are operating. At the same time, targeted studies that focus on individually marked birds or experimental manipulations often isolate mechanisms but cannot directly link their results to population and range wide consequences. Our results add to a growing number of studies that seek to make connections from individual-to-population level effects[25] in an effort to understand how the sensitivity of populations to specific conditions ultimately creates larger patterns that may differ between species. Moreover, we highlight the fact that average temperature and temperature variability are both important and that they-along with other gradients-must be considered simultaneously in order to understand the impacts of climate change. Because many climate change models predict increasingly extreme weather in addition to overall warming, one of the challenges in predicting vulnerability for bird species moving forward is to understand the extent to which average conditions versus extreme events drive demographic consequences for populations as conditions change.

## Methods

### Breeding records

We combined breeding records from three different databases. First, we requested raw breeding records from Nestwatch, a project managed by the Cornell Lab of Ornithology (www.nestwatch.org). Second, we requested raw breeding records from Project Nestwatch, a program managed by Birds Canada (www.birdscanada.org/bird-science/project-nestwatch/). Finally, we obtained publicly archived records of breeding purple martins from Project MartinWatch (www.purplemartin.org[70]). We initially manipulated each data source into a common format to allow us to merge records together. Beginning with the combined dataset, we then proceeded through a series of filtering and quality control steps to arrive at a final dataset for analysis.

While the Nestwatch database includes some older observations, most nest records are from 1995 or later (>98%); we therefore removed older records to focus on the period from 1995 to 2020. We also removed species that had fewer than 300 records. This minimum sample size was somewhat arbitrary but given the complexity of models that we planned to fit, the desire for spatial and temporal sampling, and the need to model ECEs which, by definition, only occur in a small percentage of nesting attempts, we chose a 300 nest minimum to ensure that we could reliably fit models for each species included. For a species with exactly 300 nests, we would only expect to observe an average of 7.5 nestling attempts that experienced a 2 SD cold snap or heatwave. We also removed European starling (*Sturnus vulgaris*) and house sparrow (*Passer domesticus*) records from the dataset, because these two invasive species are often considered nuisance pests and it was unclear whether a high reporting rate of failed nests reflected human eviction from nest boxes. This resulted in 24 remaining species that we considered further (Table S1).

Next, we removed any records that had incomplete location data, that were outside of the study extent, were flagged by the data providers as having errors, or that had impossible values reported (e.g., more young than eggs or negative clutch sizes). We also removed nests that did not include information on the date of laying or hatching. In cases where only a laying date was provided, we inferred approximate hatching dates for successful nests by using the typical incubation length as reported in species accounts from the Birds of the World online[71]. While fledging success was reported, exact fledging date was generally unknown, and we estimated fledging dates using the typical fledging age for a species from Birds of the World online[71].

At this point, we also roughly categorized the included species into four foraging modes based on the primary way that food is obtained during the nestling provisioning period using the species accounts in Birds of the World Online. These included aerial capture (both fly-catching and aerial insectivores), sally foraging (flying to ground from a perch after visually detecting arthropod movement), gleaning (a broad category of strategies involving active search for arthropods), and seed eating. We did not have enough total species or phylogenetic variation in foraging mode to formally analyze diet, but we discuss the possible role that these foraging styles may have had on susceptibility to cold snaps and heatwaves.

Finally, we plotted distributions of clutch sizes and hatching dates for each of the 24 species and removed records that had values that were likely due to data entry error (i.e., outside of the possible range for each species). This filtering step was somewhat subjective, but we were conservative in setting limits so that only records well outside the expected ranges were excluded. The end result of these filtering steps was a final dataset that included 301,514 breeding records from 24 species. The spatial extent of the dataset is illustrated in Figure S1 and the number of nests included for each species is shown in Table S1. Using the final set of filtered breeding records, we determined the elevation of each nest by accessing a digital elevation model raster spanning the records using the elevatr package in R[72]. Temperature data were added to each nest record as described below.

### Spatial and temperature data

**Spatial data.** We downloaded a base map for the region encompassing the breeding records included using the package rnaturalearth[73] in R version 4.0.2[74]. For some analyses (see below) we wanted to summarize breeding records and historical temperature by grouping records and weather stations that were recorded close together. To facilitate this grouping, we created a grid of hexagons each with an area of 40,000 km$^2$ across the land area of North America with the st_make_grid function from package sf[75]. We then clipped this grid to include only hexagons that contained breeding records from the dataset described above.

**Temperature records.** We accessed historical temperature data from meteorological stations in the United States using the package rnoaa[76] and in Canada using the package weathercan[77]. In both cases, we filtered stations to include those that were reported to have at least 50 years of data between 1920 and 2020 (not all stations actually yielded 50 years of data because some were active but missing the data we required). Using these criteria, we identified 2608 stations from the National Oceanic and Atmospheric Administration (NOAA) and 1125 stations from Environment and Climate Change Canada (ECCC) that covered the spatial extent of our breeding records (Fig. S2). From

these stations we downloaded all available daily high temperature records between 1920 and 2020. When summarizing temperature data for the hexagonal grid, we averaged all stations that were within the bounds of each hexagonal grid cell.

**Temperature anomalies.** We downloaded a monthly temperature anomaly raster from the Berkeley Earth project (www.berkeleyearth. org/data/). This data product compares the monthly average temperature within each 1° square grid to the average temperature over the period 1951–1980. Monthly anomalies in each year are expressed relative to that 30-year baseline period. Because we were interested in conditions during the breeding season, we extracted the anomaly for April, May, and June for each year from 1920 to 2020. We next clipped the square anomaly grid to the boundaries of each equal area hexagon and calculated a weighted average representing the anomaly within each hexagon and then merged these readings with the breeding records described above. We hereafter refer to this averaged value as the spring temperature anomaly.

**Definitions of extreme climate events.** In order to usefully summarize and analyze long-term climatic data, we had to make decisions about what counts as an extreme event. These decisions included choices about how to handle the severity, duration, and timing of cold snaps and heatwaves. We followed two connected, but distinct strategies for the historical analysis of the timing of extreme events and for the analysis of the biological effects of extreme events on each species. For historical analyses, we needed to select absolute temperature values and we defined these based on the distribution of temperatures actually experienced by the 24 species we studied (see details below).

For the analyses focused on fitness effects for individual species, we were interested in the evidence for ECEs resulting from extreme temperatures. The definition of an ECE is inconsistent[45,46], but typically these events refer to a combination of both extreme climatic conditions defined based on a distribution (e.g., events occurring <5% of the time) and a negative biological response, sometimes requiring a long recovery period[45]. No single definition is universally applicable across studies[46], but for the purposes of our study, we considered our results to indicate evidence for an ECE if 3-day cold or heat events two standard deviations from the species mean were associated with reduced reproductive success for each species.

The exact choice of a 3-day time period for considering cold snaps and heatwaves was also somewhat arbitrary, but previous work suggests that multi-day poor weather events tend to have a greater impact on offspring survival than single day events. This pattern has been observed in raptors[61,78,79], in swallows[38,80], and in other passerines[34]. We acknowledge that longer cold snaps and heatwaves likely have even more severe impacts, but they will also be correspondingly rarer to observe when matched with breeding records and 3 days represents a compromise between a multi-day challenge and enough observation of extreme temperatures to allow us to model fitness effects.

**Categorizing patterns of historical cold snap and heatwave occurrence.** Using the averaged daily high temperature for each grid cell, we determined the date of the latest 3-day cold snap and the date of the earliest 3-day heatwave for each year and hexagon cell between 1920 and 2020. For this analysis, it was necessary to use absolute temperature values to define a cold snap and heatwave so that we could ask how the annual timing of the same temperature conditions may have changed over the past 100 years. We selected the threshold temperatures used by examining the distribution of temperatures recorded during nest attempts for the 24 species included in our study.

For each species, we determined the 5th, 10th, and 20th percentile of 3-day temperatures (for cold snaps) along with the 80th, 90th, and 95th percentile of 3-day temperatures (for heatwaves) experienced during all the nesting attempts in our cleaned dataset (see below for

details on nest level temperature). This resulted in 24 temperatures for each percentile value. We then averaged these species-specific estimates to arrive at a single threshold value for each percentile (Fig. S3).

This summary resulted in cold snap thresholds of 13.9°, 15.5°, and 17.4 °C along with heatwave thresholds of 32.1°, 33.6°, and 34.8 °C. In each year, we only considered cold snaps and heatwaves that occurred after the 60th and before the 240th day of the year (approximately March 1st to August 28th), because we were interested in the timing of these events in relation to breeding activities. We present all three thresholds in most analyses and in all cases the patterns were qualitatively similar regardless of threshold, but for some summary plots at the continental scale we used the milder thresholds (17.4° and 32.1°) because these allowed for the inclusion of a wider geographic area since the most extreme temperatures for cold and heat were rarely recorded at low and high latitudes, respectively.

**Historical trends in cold snap and heatwave anomalies.** We derived a measure of yearly cold snap and heatwave timing anomaly for each grid cell to compare with the average temperature anomaly data described above. To do this we averaged the last cold snap and earliest heatwave date for the years 1951–1980 in each grid and then for each year and grid combination calculated the deviation from that average date. Negative values for the anomaly indicate years in which the last cold snap or earliest heatwave date occurred earlier than the historical average and positive values indicate years in which the last cold snap or earliest heatwave occurred later than the historical average. These values were calculated to be comparable to the temperature anomaly data described above.

**Temperatures experienced by individuals during breeding.** Using the temperature data described above, we matched each breeding record to temperature from the most similar station using a two-step process. We first matched records to the closest station. We next compared the difference in elevation between the breeding record and the station. If the closest station differed in elevation by >300 m, we searched for the station at the most similar elevation within 50 km. Using this approach, we paired records to weather stations that were 18.6 ± 10.5 km (SD) from the nest and within 47.2 ± 67.1 m elevation. While this approach resulted in the best matched station, we also note that standardized temperature records are usually recorded ~1.5 m above the ground, whereas the species included in our dataset could have nested at varying heights and in open cups or nest boxes. Thus, the exact thermal environment for nests likely differed from the temperature records we used, but we could not account for this detailed level of variation.

Using the identified station, we determined the coldest and hottest 3-day period experienced during each nesting attempt separately for the incubation and nestling provisioning period. To accomplish this, we found the sequence of 3 days with the lowest combined daytime high temperature and considered this value as the lowest short term temperature exposure. For the highest temperature, we first determined the high temperature for each day in a string of 3-days; for each group of 3 days, we considered the lowest daytime high temperature, and we then selected the string of 3-days that maximized this value (i.e., the hottest 3-day period experienced). It is important to note that these temperature determinations were continuous and, unlike the historical analysis above, they did not rely on any choice of threshold values. We used species specific timing for incubation and nestling stages to ensure that the temperatures we recorded would actually have been experienced during the reproductive attempt.

**Data analysis**
**Spring temperature and cold snap timing.** We first modeled the change over the last 100 years in spring temperature anomalies and in the timing of cold snaps and heatwaves across the spatial range of the

breeding records included in our dataset. We were interested in determining how these temperature variables have changed over time, how much regional variation there is in those patterns, and the extent to which average temperature anomalies were correlated with cold snap and heatwave timing.

We initially plotted the average temperature, cold snap, and heatwave anomalies over the past 25 years (the time period covering our breeding records) in each of the hexagon grids described above for illustration purposes. We also fit a generalized additive model (GAM) for each anomaly measure using the entire time series with the anomaly in each grid-year combination as the response variable along with a basis smoothed predictor variable for year and a spatial smooth for latitude and longitude to account for spatial autocorrelation. We used these models to describe the overall change in spring temperatures and cold snap or heatwave timings and the degree of spatial variation in those changes.

Next, we fit a GAM with the average timing of last cold snap or earliest heatwave anomaly for each grid over the past 25 years as the response variable and with the corresponding average spring temperature anomaly, and a spatial smooth for latitude and longitude as predictor variables. These models were used to infer whether spring temperature anomaly and the timing of the latest cold snap or earliest heatwave covaried, while accounting for spatial autocorrelation in the dataset.

**Impact of cold snaps and heatwaves on reproductive success.** To model the impact of cold snaps and heatwaves on reproductive success, we used a two-step approach to fit GAMs for each species separately for the incubation and nestling provisioning phase. First, as weather measurements were indicative of fine-scale regional conditions rather than those measured at specific nest locations, we calculated the response to weather conditions by averaging the number of chicks fledged grouped by hatch date and nearest weather station. We then created a model using these averaged records and 3-day temperature extremes experienced during the incubation period with mean number of chicks fledged as the response variable.

Next, we fit the same model for each species, but restricted records to those that hatched at least a single chick successfully and included only temperature records during the nestling provisioning stage. This allowed us to minimize the direct effects of temperature extremes during incubation on our models investigating the nestling phase. We separated between these two life history stages because we suspected that temperature extremes experienced during the nestling provisioning phase could have a stronger direct effect on fledging success, as the ability of the parent to buffer against extreme temperatures varies[31]. We implemented a Gaussian model for both the incubation and nestling provisioning models as they have lower false positive Type I error rates when compared to a Poisson due to overdispersion, and thus tend to be more conservative[81].

In both models, the structure of the GAM was identical. Predictors included a basis smooth for the coldest and hottest 3-day period during either reproductive stage (incubation stage or provisioning) as well as a spatial smooth for latitude and longitude to account for spatial autocorrelation. The models also included smoothed predictors for date and a random effect for year. The basis dimension value of k for the spatial smooth was chosen iteratively by comparing the effective degrees of freedom (edf) with the k-index as per Wood 2017[82]. Predictors were checked for whether their smooths contributed unique information to the model (concurvity). We fit each model separately for each of the 24 species included in our analyses. Not all species had records in every year, so the number of years included in each model varies by species, and the basis dimension value was set to this in each model. To facilitate comparison between species, we standardized the coldest and hottest 3-day period within species so that the mean was zero and standard deviation was one for each species.

We summarized sensitivity to cold snaps and heatwaves from these models by calculating pointwise estimates for number of offspring for each species during a 2 standard deviation cold snap or heatwave that occurred during either incubation or the nestling provisioning phase while controlling for the other model parameters. We then converted the estimates to model predicted relative fitness by taking the predicted number of offspring fledged divided by the average number of offspring fledged for each species. We considered point estimates with confidence intervals that did not overlap one (the average value of relative fitness) to indicate significant sensitivity to temperature extremes.

Species that experienced these reductions in fitness would be considered to have experienced an ECE based on the combined climatic extremity and biological response definition outlined by Smith, 2011[45]. However, Bailey & van de Pol (2016) argue that using an arbitrary distribution cutoff might miss important biological responses at different values or when responses differ between species[46] (in our case using 2 SD results in point estimates at approximately the 2.5th and 97.5th percentiles). In an online appendix, we include the full relative fitness surface for each species across the range of coldest and hottest 3-day periods. These comparisons do not rely on any arbitrary choice of a single point estimate from the climatic distribution.

**Latitudinal variation in cold snap and heatwave susceptibility**
After fitting the global models described above, we investigated whether the susceptibility to cold snaps and heatwaves varied from the northern to southern limits of the breeding range in three species: purple martins, eastern bluebirds, and tree swallows. We chose these three species because they had the largest sample sizes and because all three also have a wide latitudinal distribution and the global analysis above indicated susceptibility to both cold snaps and heatwaves. For each species, we split the records into 5 latitude bands with an equal number of nest records per band and then repeated the global analyses exactly as described above within each band. The number of records was sufficient for these species that even after splitting into five datasets each latitude band included >9,000 nesting records for each species. For tree swallows, we limited this analysis to records east of the Rocky Mountains because mountain and western populations have very different breeding timing from eastern populations at similar latitudes.

**Reporting summary**
Further information on research design is available in the Nature Portfolio Reporting Summary linked to this article.

# Data availability

The nesting data used in this study were obtained from three citizen science databases and can be retrieved from each of them. Both the NestWatch program and Project Nestwatch have data access portals (https://nestwatch.org/nw/public/export and https://www.birdscanada.org/bird-science/project-nestwatch). Access to data can be obtained by submitting a request and agreeing to data use policies. Data from Project Martinwatch are publicly archived on Dryad (https://doi.org/10.5061/dryad.msbcc2fwq). Historical temperature data were accessed from the Berkeley Earth project (www.berkeleyearth.org/data/) or from meteorological station records maintained by NOAA in the United States (https://www.ncei.noaa.gov/cdo-web/) and by ECCC in Canada (https://climate.weather.gc.ca/historical_data/search_historic_data_e.html).

# Code availability

All data processing, analyses, and figures were created in R version 4.0.2. A complete set of annotated code to reproduce the full analysis, manuscript, and supplemental materials is permanently archived on Zenodo (https://doi.org/10.5281/zenodo.8208669). We used the following R packages in our analysis: tidyverse v1.3.0; rnoaa v1.3.6.94;

sf v0.9-8; rnaturalearth v0.1.0; ggpubr v0.4.0; data.table v1.14.2; weathercan v0.5.0; elevatr v0.3.4; mgcv v1.8-31; and dplyr v1.0.8. The archived repository also includes an appendix with complete model summary tables for every GAM described in the results. Figure S4 provides a conceptual overview of the complete analysis pipeline with reference to each script in the code repository.

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

## Acknowledgements
We would like to thank the NestWatch program, Project NestWatch, and Project MartinWatch for creating and maintaining the databases that made these analyses possible. We thank Robyn Bailey for help in accessing these records and guidance on working with these data. Maren Vitousek and the Vitousek lab provided helpful discussions and feedback on the project. We also thank the many community members

who collected and submitted data. CCT was supported by National Science Foundation Grant #2128337.

## Author contributions

C.C.T. & J.R.S. contributed equally to conception and analysis of the manuscript. C.C.T. drafted the paper with input from J.R.S. and both authors contributed to revisions.

## Competing interests

The authors declare no competing interests.
