## [Peer Review File · Nature Communications]

Inconsistent shifts in warming and temperature variability are linked to reduced avian fitnessREVIEWER COMMENTS

Reviewer #1 (Remarks to the Author):

With pleasure I have reviewed the manuscript titled "Inconsistent shifts in warming and temperature variability negatively impact avian fitness" submitted for the journal *Nature Communications*. The authors present a continental-scale analysis of long-term nesting data of North American birds to assess how climate warming trends and spatio-temporal variability in climate anomalies influence fitness for 24 species. The authors present an impressive collation of community science data sets to approach an individual-to-population response on how these climate signatures may be impacting avian species from a fecundity perspective.

I think the authors would do well to think about their approach to extreme climatic events and how they are empirically defining them in this study. How one defines an extreme climatic event is critical to analyses and interpretations of climate-based studies and thus these definitions are not purely semantics. For instance, there are both climatological and biological definitions of ECEs. ECEs are events where conditions occurring are rare relative to the -distribution- of those conditions (i.e., occurring <5% of the time). In addition, ECEs must also result in an extreme negative biological response for an organism (constraining a response parameter to the lower tail of the parameter's distribution). I challenge the authors to think about these definitions within the framework of their study and be explicit about how their work may fall within these proposed synthetic definitions and mechanistic frameworks. For further reference, please see:

Smith, M. D. 2011. An ecological perspective on extreme climatic events: a synthetic definition and framework to guide future research. *Journal of Ecology* 99:656-663.
Bailey, L., and van de Pol, M. 2016. Tackling extremes: challenges for ecological and evolutionary research on extreme climatic events. *Journal of Animal Ecology* 85:85-96.

My other concern is tied to the above thought: selection of cold-snap and heat-wave thresholds. I appreciate that the authors explored multiple thresholds for both metrics, and provided at least one justification for the 18.5C threshold for cold-snaps. This is such a critical part of your analyses and interpretations that generalizing such thresholds across all species was a challenge for me. For instance, you 18.5C cold threshold is determined from a source that strictly based on tree swallows for a site in New York. Yet, you infer this relationship across species and spatial domain. You select 26C as your upper threshold, yet provide no support for this value (unless I overlooked it). Perhaps this is justified given that there seems to be no significant differences across heat wave thresholds based on GAMS, and what seems to be only slight significance across cold thresholds based on GAMS (unclear as no statistical values were provided and the authors refer to these relationships as "not consistently different" or "tending to occur early"). I appreciate that a threshold needs to be set for your analysis to move forward. How does this threshold fit within the mechanistic framework proposed for defining ECEs? This would help produce some standardization and comparability for your study moving forward.

I understand that this latter point was likely given much thought by the authors during the design of their study. At a minimum, I believe they can provide more or better justification beyond a single tree swallow study or an uncited upper threshold that may not require any additional analyses.

Overall, I believe the manuscript offers valuable information toward understanding how climate variability is driving individualistic responses and likely population-level performance at an impressive spatial and temporal scale. Beyond what was mentioned above, I have a few other minor points to consider below. I appreciate the work that has been put into this study by the authors and for the opportunity to review it.

INTRODUCTION

Line 44: Would be worth mentioning this wasn't the case with wetland birds (i.e., presumably waterfowl). This offers some unique perspectives to consider within the auspices of climate change (i.e., how does temperature influence soil moisture (decreased water viscosity) and thus wetland inundation under future climate scenarios?). Are we seeing wetland bird increases due to very targeted and successful adaptive management strategies, yet these gains could be offset by these ECEs?

Lines 50-55: Another reason average temperature alone is not sufficient: disparate warming rates between diurnal and nocturnal periods. It's the flaw of averages, something which I think this paper makes an important point about, but it doesn't just happen within the context of ECEs. This could be a good opportunity to briefly introduce the idea of direct and indirect effects of these climate changes on your model taxonomic group. There are obvious direct physiological implications, you hint at indirect effects (decreased insect availability for foraging) (Line 65), but there are other empirical examples that could help build your narrative. For instance, DeGregorio et al. 2015 (Indirect effect of climate change: shifts in ratsnake behavior alter intensity and timing of avian nest predation [Ecological Modelling 312:239-246]) provides some thoughts on disparate warming patterns across the diel cycle for nesting avian species. Just something to consider.

Line 60: Life history traits are just one aspect though. What about functional traits? Kearney et al. 2021 Where do functional traits come from? The role of theory and models in Functional Ecology offer a nice framework that have direct implications to the narrative you are building for interspecific and possibly even intraspecific variation to climate change responses.

Line 65: It would be helpful to describe the mechanism of reduced insect availability here for non-entomologists. Straight mortality? Decreased activity and thus reduced foraging success?

METHODS

Line 116: Why 300? Was this arbitrary, based on previous work, or based on the distribution of data points by species? Also, why this temporal extent? Your choice of temporal extent will have direct implications on your climate signatures. Are these dates specific to the community science programs you pulled from? If so, make that explicit.

Line 124: Please change "study range" to "study extent".

Line 165: How did you account for the spatial mismatch in your 2degree versus 1degree grid here. Also, was this 1degree grid square as opposed to hexagonal? A brief mention of how you addressed this would be helpful.

Line 177: This is a huge assumption of your analyses and based on a single study of tree swallow. This ignores interspecific variations but more importantly ignores ecotypic adaptation. I appreciate that you had to select a threshold and that you wanted to make it biologically meaningful/justified. At a minimum this needs to be discussed later, but more importantly how did you select your alternative thresholds? ECEs can be defined a lot of different ways (see above), and one way is to have an event that is extreme relative to the distribution of conditions, rather than extreme in a way that elicits an extreme biological response. Where did these alternative thresholds fall with regards to ECEs?

Line 201: This may be semantics but generally weather stations are standardized at 1.5m AGL, while avian nesting varies from ground level to tree crowns. Moreover, many of the species in your dataset are assumingly records from nest boxes/artificial colony structures (likely varying in height as well). I appreciate you trying to standardize these two data points based on digital elevation models, but a mention of this within-dataset variation should be made.

Line 246 and 248: You should maintain consistency in the terminology of chicks and nestlings here as I assume they mean the same thing.

RESULTS

Figure 1: It would help to provide a bit more information on temperature anomaly calculation, and cold/heat anomaly calculations in the figure caption to make it stand alone. For instance, you say "Cold snap anomaly is the difference in timing...". Difference in timing to what specifically? I know this is in your methods but just thinking about how you can better make this figure stand alone.

Line 317: Please specify the Figures you are referring to. There seems to be inconsistency in what should be figure 2? Could you please correct this.

Figure 2: Please specify these are data from North America spatial domain to ensure figure stands alone.

Figure 3: Please specify these are data from North America spatial domain to ensure figure stands alone.

Figure 4: It would be of value to label what the latitude bands are for figures 4B-D in the Y axis.

DISCUSSION

Lines 455-460: I think this is a good discussion on the differences observed between responses to cold-snaps versus heat-waves. However, I think there is more that could be added. Your mechanism of response to cold-snaps focused on forage availability, while your mechanism of response to heat-waves focused on physiological limitations. Yet, avifauna have shown to

behaviorally thermoregulate and exchange possible foraging opportunities to mitigate risks of hyperthermia. Moreover, many species of avifauna have been shown to select nesting sites that reduce exposure to high operative (or black body) temperatures at the microsite level when compared to the available landscape. There could be an issue of scale mismatch that may be driving the weaker responses to heat waves beyond just you selecting a longer period than 3 days. I think you could build your narrative here accordingly.

Reviewer #2 (Remarks to the Author):

This study uses a large dataset of avian breeding records from multiple species across North America to investigate trends in the timing of extreme climatic events (ECEs) and the impacts of ECEs on reproductive success. The authors revealed that despite increasing average spring temperatures, there was no clear change in the timing and/or magnitude of either cold snaps or heat waves. They also showed that both heat waves and cold snaps had negative impacts on fitness, but that these effects varied by species and were generally more pronounced for cold snaps. Finally, by comparing the effects of ECEs on reproductive success across a latitudinal gradient for three species, the authors found some evidence that the effects of cold snaps and heat waves were more pronounced at northern and southern range limits, respectively. The authors suggest that this may point to ECE impacts contributing to range shifts. I enjoyed reading the manuscript – the topic is of great interest within ecology and the results build on our understanding of the impacts of ECEs and their context specificity. I also felt that the analyses and associated interpretations were largely appropriate and that the manuscript is well written. However, I have some comments on the analyses and some suggestions to aid clarity:

- I have some queries about the ECE definitions and suggest that the justification needs to be more explicit in the Methods section. For example, it was unclear to me why a 3-day timespan was picked, why the cold snap definition was based on aerial insectivore food availability given the use of both aerial insectivores and species with other diets/foraging modes, and why 26 degrees was selected to denote a heatwave. I appreciate that you also used alternate ECE definitions in supplementary analyses, but I still think it's important to be clear on why you selected the values you did.
- Related to the above point, I was unclear on why the ECE definition was not carried through to the fitness analyses. The lack of congruence between these analyses makes it difficult to link the results between the sets of analyses and also makes it difficult (as the authors acknowledge) to draw comparisons between species. Was this difference in approach due to the ECE definition in the historical analyses being too strict and thus a lack of ECEs in many of the periods? I think keeping the ECE definition consistent throughout would make for a stronger overall paper, but if there was a clear reason why this was not possible, it should be stated in the Methods.
- Given that drawing comparisons between species with particular diets and foraging modes proves important, particularly for points made in the Discussion, it would be helpful to see these distinctions made in the Methods and carried throughout, for example by grouping species in the figures or something along those lines.
- Line 116 – why was 300 breeding records used as the cut-off and how many species did this exclude?
- Line 365 – your analysis currently doesn't account for the grouping of records by weather station and thus the potential similarity in these records as a result of shared conditions. This currently won't be well accounted for by the modelled spatial autocorrelation, so should be added as an additional random effect.
- There appear to be a few LaTeX-associated issues throughout (e.g., Figure not referred to correctly on lines 217 and 219, zenodo link missing on line 205)

- Line 294 – I don't think your results demonstrate that exposure to temperature variability could increase with climate change given that you don't look at the trends in variability, but rather at trends in the timing of ECEs. Therefore, I think you should tone down this point and ensure your conclusions are supported by specific results from your analyses.

Reviewer #3 (Remarks to the Author):

The manuscript by Taff and Shipley focuses on examining the potential impacts of cold snaps and heat waves on the North American breeding bird population. The authors have employed a combination of datasets, including local nesting and hatching success data from citizen science initiatives, as well as historical weather records from meteorological stations. This approach allows them to investigate the effects of climatic extremes on population fitness over the past few decades.

The findings of the study indicate that temperature extremes have a detrimental impact on the fitness of approximately half to two-thirds of the species studied. Additionally, for widely distributed species, the authors demonstrate that the effects of these extremes vary based on latitude. By analyzing the occurrence of extreme weather events during the breeding season, the authors reveal that while average spring temperatures have consistently increased throughout North America over time, the patterns of cold snaps and heat waves exhibit more complex dynamics.

In conclusion, the authors emphasize the necessity of acquiring detailed knowledge regarding species' sensitivity to extreme weather events to accurately predict population responses to ongoing climate change.

The manuscript presents valuable and scarce information regarding the link between individual responses to climatic events and population sensitivity. The authors' methodological approach is innovative, straightforward, and has the potential to establish a new standard in the field. The manuscript is well-written and comprehensive. I firmly believe that its publication in Nature Communications will garner the appropriate attention and receive a positive response from the research community.

There are a few suggestions that could enhance the manuscript further.

1. Authors state that all the code will be available alongside with the manuscript, but at the moment is it not, so I cannot see how well it is developed and how easy it would be to re-use it.
2. The methodological details, specifically model equations, are not presented in the text and are sometimes hard to follow.
3. I feel that spatially explicit connection between individual and population level could have been developed better. For example authors could map average fitness in species' ranges in the past and now to show where the effects seem to be stronger. They could also compare these maps between the species. Authors could also have looked into spatially explicit predictions of the occurrence of the extreme events and use these as to predict future population fitness. These changes are not necessary, but would nicely conclude the manuscript and would connect the parts with the fitness and spatial maps of extreme events.
4. I think that presentation of fitness effects of extremes followed by their spatial distribution would improve the manuscript flow.

Please find a few minor questions and comments below.

L144 - Are you utilizing Natural Earth Data for any analysis? If not, it may not be necessary to mention it here.

L147 - Consider choosing equal area hexagons instead of 2-degree intervals. The former would facilitate easier comparisons and extrapolation.

L225-232 - It would greatly enhance the manuscript to include a schematic of the analysis or, at the very least, provide some equations used.

L234 - It is worth noting that the first and last events may not always follow a normal distribution. Therefore, it is essential to perform checks on the error distribution before utilizing gams with Gaussian errors.

L249 - Could you clarify the error distribution used in this particular case?

L258-259 - While Poisson models are susceptible to over-dispersion, it would be helpful to know if you encountered this issue. Additionally, incorporating individual random effects can address this problem. It is also more common to switch to a negative-binomial distribution rather than a Gaussian one.

L307 - In Figure 1, consider using different color schemes for the three maps. Alternatively, if using a single palette, color them based on their effects on birds, such as assigning the same positive color to early cold snaps and later heat waves.

L307 - Furthermore, it would be interesting to include the duration of the average breeding season (between the last cold snap and the first heat wave) in Figure 1. This could provide insights into whether the duration becomes critically short in certain locations.

L317 - There appears to be an issue with the figure at this point (no figure number and no figure caption).

1. General comments

We would like to thank the reviewers for an exceptionally generous and thoughtful set of reviews. We really appreciate the care put into the reviews and we think that the manuscript has been greatly improved through this revision process.

We have thoroughly revised our initial submission and included detailed responses that address all the points raised by the referees below. With the new submission, we also include a full set of code to reproduce analyses and figures presented in the text. In addition to the specific changes listed below, we made a number of small changes to the text to improve clarity or expand on points brought up by the reviewers and we have provided a version with all changes marked to highlight these differences.

Comments from Reviewer 1:

2. With pleasure I have reviewed the manuscript titled “Inconsistent shifts in warming and temperature variability negatively impact avian fitness” submitted for the journal Nature Communications. The authors present a continental-scale analysis of long-term nesting data of North American birds to assess how climate warming trends and spatio-temporal variability in climate anomalies influence fitness for 24 species. The authors present an impressive collation of community science data sets to approach an individual-to-population response on how these climate signatures may be impacting avian species from a fecundity perspective.

Thank you for the kind and thoughtful review of our manuscript. We have responded in detail to each comment below.

3. I think the authors would do well to think about their approach to extreme climatic events and how they are empirically defining them in this study. How one defines an extreme climatic event is critical to analyses and interpretations of climate-based studies and thus these definitions are not purely semantics. For instance, there are both climatological and biological definitions of ECEs. ECEs are events where conditions occurring are rare relative to the -distribution- of those conditions (i.e., occurring <5% of the time). In addition, ECEs must also result in an extreme negative biological response for an organism (constraining a response parameter to the lower tail of the parameter’s distribution). I challenge the authors to think about these definitions within the framework of their study

and be explicit about how their work may fall within these proposed synthetic definitions and mechanistic frameworks. For further reference, please see:

Smith, M. D. 2011. An ecological perspective on extreme climatic events: a synthetic definition and framework to guide future research. *Journal of Ecology* 99:656-663.

Bailey, L., and van de Pol, M. 2016. Tackling extremes: challenges for ecological and evolutionary research on extreme climatic events. *Journal of Animal Ecology* 85:85-96.

Thank you for pointing us to these references. The papers suggested are really helpful in refining our thinking and we agree that we were not explicit or careful enough in defining ECEs in the initial submission. We have carefully edited the revision so that our terminology is more precise. For example, in the introduction and abstract we now mostly refer to extreme hot or cold temperatures rather than ECEs when we were not making an explicit connection to biological effects.

We have also added several paragraphs to the methods that outline and clarify our approach to ECEs. Terminology changes are included throughout the methods, but the most significant change is this additional sub section:

“Defining cold-snaps, heat-waves, and ECEs. In order to usefully summarize and analyze long-term climatic data, we had to make decisions about what counts as an extreme event. These decisions included choices about how to handle the severity, duration, and timing of cold-snaps and heat-waves. We followed two connected, but distinct strategies for the historical analysis of the timing of extreme events and for the analysis of the biological effects of extreme events on each species. For historical analyses, we needed to select absolute temperature values and we defined these based on the distribution of temperatures actually experienced by the 24 species we studied (see details below).

For the analyses focused on fitness effects for individual species, we were interested in the evidence for ECEs resulting from extreme temperatures. The definition of an ECE is inconsistent⁴⁸, but typically these events refer to a combination of both extreme climatic conditions defined based on a distribution (e.g., events occurring <5% of the time) and a negative biological response, sometimes requiring a long recovery period^{48,49,50}. No single definition is universally applicable across studies⁴⁸, but for the purposes of our study, we considered our results to indicate evidence for an ECE if 3-day cold or heat events two standard deviations from the

species mean were associated with reduced reproductive success for each species.”

And also this additional paragraph in the data analysis section:

“Species that experienced these reductions in fitness would be considered to have experienced an ECE based on the combined climatic extremity and biological response definition outlined by Smith⁴⁹. However, Bailey & van de Pol⁴⁸ argue that using an arbitrary distribution cutoff might miss important biological responses at different values or when responses differ between species (in our case using 2 SD results in point estimates at approximately the 2.5th and 97.5th percentiles). In an online appendix, we include the full relative fitness surface for each species across the range of coldest and hottest 3-day periods. These comparisons do not rely on any arbitrary choice of extremity from the climatic distribution for point estimates.”

4. My other concern is tied to the above thought: selection of cold-snap and heat-wave thresholds. I appreciate that the authors explored multiple thresholds for both metrics, and provided at least one justification for the 18.5C threshold for cold-snaps. This is such a critical part of your analyses and interpretations that generalizing such thresholds across all species was a challenge for me. For instance, you 18.5C cold threshold is determined from a source that strictly based on tree swallows for a site in New York. Yet, you infer this relationship across species and spatial domain. You select 26C as your upper threshold, yet provide no support for this value (unless I overlooked it). Perhaps this is justified given that there seems to be no significant differences across heat wave thresholds based on GAMs, and what seems to be only slight significance across cold thresholds based on GAMs (unclear as no statistical values were provided and the authors refer to these relationships as “not consistently different” or “tending to occur early”). I appreciate that a threshold needs to be set for your analysis to move forward. How does this threshold fit within the mechanistic framework proposed for defining ECEs? This would help produce some standardization and comparability for your study moving forward. I understand that this latter point was likely given much thought by the authors during the design of their study. At a minimum, I believe they can provide more or better justification beyond a single tree swallow study or an uncited upper threshold that may not require any additional analyses.

Thank you for the comment. This was unclear and not well justified in our initial submission (as all three reviewers pointed out!). We think there are two separate issues here.

First, there is the issue of how much a choice of threshold may have influenced our results. We think it was not clear in the first submission that the threshold was only used in the historical analysis, but did not apply at all to the main analysis of relative fitness on nests. That analysis used a continuous measure of temperature for each nest and so did not rely on any threshold value. The summary figure (fig 4) does impose a distributional threshold by plotting point estimates at 2 SD, but we now also include an online appendix that includes the full fitness surface for each species. We have also made it much clearer in the revised methods how and why these two analyses differ (see response to comment #3 above).

Second, there is the issue of how the actual threshold values for the historical analysis were chosen. We agree that this was not well justified in the previous version. We have completely re-worked those analyses so that the new thresholds are based on the actual distribution of experienced temperatures during nest attempts combined over all of the included species. The results are qualitatively similar, but we think this is a much clearer and less arbitrary justification. We explain this in the revised methods and include a new supplemental figure detailing the choice of thresholds. Here is the new text:

“For this analysis, it was necessary to use absolute temperature values to define a cold-snap and heat-wave so that we could ask how the annual timing of the same temperature conditions may have changed over the past 100 years. We selected the threshold temperatures used by examining the distribution of temperatures recorded during nest attempts for the 24 species included in our study.

For each species, we determined the 5th, 10th, and 20th percentile of 3-day temperatures (for cold-snaps) along with the 80th, 90th, and 95th percentile of 3-day temperatures (for heat-waves) experienced during all the nesting attempts in our cleaned dataset (see below for details on nest level temperature). This resulted in 24 temperatures for each percentile value. We then averaged these species-specific estimates to arrive at a single threshold value for each percentile (Figure S3).

This summary resulted in cold-snap thresholds of 13.9, 15.5, and 17.4 C along with heat-wave thresholds of 32.1, 33.6, and 34.8 C. In each year, we only considered cold-snaps and heat-waves that occurred after the 60th

and before the 240th day of the year (approximately March 1st to August 28th), because we were interested in the timing of these events in relation to breeding activities. We present all three thresholds in most analyses and in all cases the patterns were qualitatively similar regardless of threshold, but for some summary plots at the continental scale we used the milder thresholds (17.4 and 32.1) because these allowed for the inclusion of a wider geographic area since the most extreme temperatures for cold and heat were rarely recorded at low and high latitudes, respectively.”

5. Overall, I believe the manuscript offers valuable information toward understanding how climate variability is driving individualistic responses and likely population-level performance at an impressive spatial and temporal scale. Beyond what was mentioned above, I have a few other minor points to consider below. I appreciate the work that has been put into this study by the authors and for the opportunity to review it.

Thank you.

6. Line 44: Would be worth mentioning this wasn't the case with wetland birds (i.e., presumably waterfowl). This offers some unique perspectives to consider within the auspices of climate change (i.e., how does temperature influence soil moisture (decreased water viscosity) and thus wetland inundation under future climate scenarios?). Are we seeing wetland bird increases due to very targeted and successful adaptive management strategies, yet these gains could be offset by these ECEs?

We have added a brief mention here of this idea. We certainly agree that this is an important point. Since we don't have breeding data for wetland birds (particularly waterfowl) in the community science datasets we used here, we don't want to devote too much space to this. But we agree that it would be very interesting to do a similar analysis to what we present using some of those species that have not been as clearly impacted by climate change (at least in terms of their total population sizes). Here is the revised section:

“However, the declines in overall abundance are far from uniform, with some groups—such as aerial insectivores—showing steep declines while others—such as many wetland birds—have not declined or have even increased.¹⁶ These differences suggest that life history and habitat

requirements many play an important role in determining how different species respond to climate change.”

7. Lines 50-55: Another reason average temperature alone is not sufficient: disparate warming rates between diurnal and nocturnal periods. It's the flaw of averages, something which I think this paper makes an important point about, but it doesn't just happen within the context of ECEs. This could be a good opportunity to briefly introduce the idea of direct and indirect effects of these climate changes on your model taxonomic group. There are obvious direct physiological implications, you hint at indirect effects (decreased insect availability for foraging) (Line 65), but there are other empirical examples that could help build your narrative. For instance, DeGregorio et al. 2015 (Indirect effect of climate change: shifts in ratsnake behavior alter intensity and timing of avian nest predation [Ecological Modelling 312:239-246]) provides some thoughts on disparate warming patterns across the diel cycle for nesting avian species. Just something to consider.

This is a great point and we have expanded the discussion of this idea in the introduction (both the discussion of mean vs. average temperature and the idea of indirect and direct effects citing the paper suggested here). We have modified the language around this point throughout the intro, but the most relevant addition is:

“However, it is increasingly apparent that changes in mean temperatures are insufficient to understand the response to climate change, as it only provides a simple approximation of changes in minimum (nighttime) and maximum (daytime) temperatures which can change independently in pace, magnitude, and variability . Different organismal traits such as timing of reproduction or when a species is active likely shape the response because animals may need to balance changing temperatures with other abiotic gradients, such as day length, UV radiation, or water availability.”^{28,29}

8. Line 60: Life history traits are just one aspect though. What about functional traits? Kearney et al. 2021 Where do functional traits come from? The role of theory and models in Functional Ecology offer a nice framework that have direct implications to the narrative you are building for interspecific and possibly even intraspecific variation to climate change responses.

Thanks for this suggestion. We have added a brief reference to the role of functional traits in the fourth paragraph of the introduction:

“The strength of this relationship is determined by how efficiently organisms transfer heat and energy in different environmental conditions and how these traits influence individual performance as well as the ability to survive, develop, grow, and reproduce (Kearney et al 2021).”

9. Line 65: It would be helpful to describe the mechanism of reduced insect availability here for non-entomologists. Straight mortality? Decreased activity and thus reduced foraging success?

Previous research has suggested the primary response of low temperature events is reduced insect activity and delayed emergence. Birds that rely on insect activity for detection are particularly affected, by reducing foraging success that limits their ability to provision food to their offspring during periods of high food demand (e.g., Winkler et al. 2013). We’ve edited the paragraph to describe this more explicitly:

“The vast majority of birds are reliant on insects to provide energy and nutrients to their rapidly developing offspring. For example, obligate aerial insectivores depend on flying insects and modest decreases in daytime temperatures can reduce insect activity and thereby overall foraging success³⁵ (hereafter “cold-snaps”). If these reductions in insect activity occur during periods when nestlings are most vulnerable, they can trigger widespread episodes of nestling mortality due to starvation.^{29”}

10. Line 116: Why 300? Was this arbitrary, based on previous work, or based on the distribution of data points by species? Also, why this temporal extent? Your choice of temporal extent will have direct implications on your climate signatures. Are these dates specific to the community science programs you pulled from? If so, make that explicit.

We have added a justification for the temporal extent and the choice of 300 as a minimum sample size. The temporal extent is indeed defined by what is available in the community science programs and the sample size minimum, while admittedly somewhat arbitrary, was chosen to ensure that we could reasonably expect to fit the models we planned for each species. Here is the text added to the revision describing these justifications:

“While the Nestwatch database includes some older observations, most nest records are from 1995 or later (>98%); we therefore removed older records to focus on the period from 1995 to 2020. We also removed species that had fewer than 300 records. This minimum sample size was somewhat arbitrary, but given the complexity of models that we planned to fit, the desire for spatial and temporal sampling, and the need to model ECEs which, by definition, only occur in a small percentage of nesting attempts, we chose a 300 nest minimum to ensure that we could reliably fit models for each species included. For a species with exactly 300 nests, we would only expect to observe an average of 7.5 nestling attempts that experienced a 2 SD cold-snap or heat-wave.”

11. Line 124: Please change “study range” to “study extent”.

We have made this change.

12. Line 165: How did you account for the spatial mismatch in your 2degree versus 1degree grid here. Also, was this 1degree grid square as opposed to hexagonal? A brief mention of how you addressed this would be helpful.

We used standard raster processing functions to clip the square grid to the borders of the hexagon polygons and calculated a weighted average for each hexagon. For example, if a hexagon completely covered one square and covered one half of two other squares, the value for the hexagon would be calculated as: $(\text{square1_value} + 0.5 \times \text{square2_value} + 0.5 \times \text{square3_value}) / 2$. This is a standard GIS processing technique for summarizing rasters (here temperature anomaly) within the bounds of a vector layer (here our hexagonal grid). We have added a brief description of the process to the manuscript:

“Because we were interested in conditions during the breeding season, we extracted the anomaly for April, May, and June for each year from 1920-2020. We next clipped the square anomaly grid to the boundaries of each equal area hexagon and calculated a weighted average representing the anomaly within each hexagon and then merged these readings with the breeding records described above.”

13. Line 177: This is a huge assumption of your analyses and based on a single study of tree swallow. This ignores interspecific variations but more importantly ignores ecotypic adaptation. I appreciate that you had to select a threshold and that you wanted to make it biologically meaningful/justified. At a minimum this

needs to be discussed later, but more importantly how did you select your alternative thresholds? ECEs can be defined a lot of different ways (see above), and one way is to have an event that is extreme relative to the distribution of conditions, rather than extreme in a way that elicits an extreme biological response. Where did these alternative thresholds fall with regards to ECEs?

Please see detailed responses to comments #3 and #4 above, which detail changes to deal with these points.

14. Line 201: This may be semantics but generally weather stations are standardized at 1.5m AGL, while avian nesting varies from ground level to tree crowns. Moreover, many of the species in your dataset are assumingly records from nest boxes/artificial colony structures (likely varying in height as well). I appreciate you trying to standardize these two data points based on digital elevation models, but a mention of this within-dataset variation should be made.

This is a good point. Of course we cannot address this statistically since we do not have temperature recorded directly at each nest or complete data on the height of nests, but we have added the caveat to the methods that our temperature records do not perfectly align with nest locations:

“While this approach resulted in the best matched station, we also note that standardized temperature records are usually recorded ~1.5m above the ground, whereas the species included in our dataset could have nested at varying heights and in open cups or nest boxes. Thus, the exact thermal environment for nests likely differed from the temperature records we used, but we could not account for this detailed level of variation.”

15. Line 246 and 248: You should maintain consistency in the terminology of chicks and nestlings here as I assume they mean the same thing.

We have made this change.

16. Figure 1: It would help to provide a bit more information on temperature anomaly calculation, and cold/heat anomaly calculations in the figure caption to make it stand alone. For instance, you say “Cold snap anomaly is the difference in timing...”. Difference in timing to what specifically? I know this is in your methods but just thinking about how you can better make this figure stand alone.

We have added these details to the figure caption, which now reads:

“Spring temperature anomaly (A), cold-snap anomaly (B), and heat-wave anomaly (C) for the North American spatial extent studied over the years 1995-2020. Spring temperature anomaly is defined as the average of April, May, and June using data from the Berkeley Earth Project measured in degrees C compared to the baseline period of 1950-1980. Cold-snap anomaly is the difference in average timing of the latest three day period in which temperatures did not rise above 17.4 C from 1995-2020 compared to the average date of the latest cold-snap from 1950-1980, measured in days. Heat-wave anomaly is the difference in timing of the earliest three day period in which temperatures always rose above 32.1 C from 1995-2020 compared to the average date of the earliest heat-wave from 1950-1980, measured in days. Gray cells indicate regions with missing data.”

17. Line 317: Please specify the Figures you are referring to. There seems to be inconsistency in what should be figure 2? Could you please correct this.

We apologize for the confusion here, the automatic figure numbering had a mistake that caused it to not render correctly. We have fixed it in the revision.

18. Figure 2: Please specify these are data from North America spatial domain to ensure figure stands alone.

We have added this to the figure caption.

19. Figure 3: Please specify these are data from North America spatial domain to ensure figure stands alone.

We have added this to the figure caption.

20. Figure 4: It would be of value to label what the latitude bands are for figures 4B-D in the Y axis.

We have added the latitude bands to the figures (figure 4 referenced here and also the two supplemental figures that have the same format) and to the table captions that describe this analysis.

21. Lines 455-460: I think this is a good discussion on the differences observed

between responses to cold-snaps versus heat-waves. However, I think there is more that could be added. Your mechanism of response to cold-snaps focused on forage availability, while your mechanism of response to heat-waves focused on physiological limitations. Yet, avifauna have shown to behaviorally thermoregulate and exchange possible foraging opportunities to mitigate risks of hyperthermia. Moreover, many species of avifauna have been shown to select nesting sites that reduce exposure to high operative (or black body) temperatures at the microsite level when compared to the available landscape. There could be an issue of scale mismatch that may be driving the weaker responses to heat waves beyond just you selecting a longer period than 3 days. I think you could build your narrative here accordingly.

This is a good point and we have added this sentence to the paragraph:

“One reason for these smaller effects might be that the species in our study were able to mitigate the risks of hyperthermia by changing behaviors or by choosing nesting sites that were cooler than the overall temperature recorded from nearby weather stations.”

Comments from Reviewer 2:

22. This study uses a large dataset of avian breeding records from multiple species across North America to investigate trends in the timing of extreme climatic events (ECEs) and the impacts of ECEs on reproductive success. The authors revealed that despite increasing average spring temperatures, there was no clear change in the timing and/or magnitude of either cold snaps or heat waves. They also showed that both heat waves and cold snaps had negative impacts on fitness, but that these effects varied by species and were generally more pronounced for cold snaps. Finally, by comparing the effects of ECEs on reproductive success across a latitudinal gradient for three species, the authors found some evidence that the effects of cold snaps and heat waves were more pronounced at northern and southern range limits, respectively. The authors suggest that this may point to ECE impacts contributing to range shifts. I enjoyed reading the manuscript – the topic is of great interest within ecology and the results build on our understanding of the impacts of ECEs and their context specificity. I also felt that the analyses and associated interpretations were largely appropriate and that the manuscript is well written. However, I have some comments on the analyses and some suggestions to aid clarity:

Thank you for the thoughtful review. We have responded to specific comments in detail below.

23. • I have some queries about the ECE definitions and suggest that the justification needs to be more explicit in the Methods section. For example, it was unclear to me why a 3-day timespan was picked, why the cold snap definition was based on aerial insectivore food availability given the use of both aerial insectivores and species with other diets/foraging modes, and why 26 degrees was selected to denote a heatwave. I appreciate that you also used alternate ECE definitions in supplementary analyses, but I still think it's important to be clear on why you selected the values you did.

Please see extensive responses above to comments #3 & #4 from reviewer 1, which raise these same concerns about ECE definitions and choice of thresholds. We have thoroughly revised the new submission to address these issues.

Although our exact choice of a 3-day timespan was somewhat arbitrary, previous evidence from the literature has demonstrated that multi-day exposure to poor weather events tend to have a greater impact on offspring survival than single day events. This pattern has been observed in raptors (Fisher et al. 2015, McDonald et al. 2004, Corregidor-Castro et al. 2023), in swallows (Brown and Brown 1998, Winkler et al. 2013), and in other passerines (Regan and Sheldon 2023). From our perspective, using a multi-day threshold provides a more conservative estimate in modeling the relationships between aberrant weather events than shorter thresholds; as birds exposed to shorter term events such as single day heat waves would be included as background mortality with our approach (meaning that the effects of 3-day events would need to rise above this background to be detected). Of course it is possible that an even longer threshold would have detected stronger patterns, but we had to make some choices in order to proceed with the analyses, and we include this caveat in the discussion, especially with respect to heat waves. See our related response to reviewer comment #3 above. We have added some of this detail to the methods:

“The exact choice of a 3-day time period for considering cold-snaps and heat-waves was also somewhat arbitrary, but previous work suggests that multi-day poor weather events tend to have a greater impact on offspring survival than single day events. This pattern has been observed in raptors,⁵⁵⁻⁵⁷ in swallows,^{38,58} and in other passerines.³⁴ We acknowledge that longer cold-snaps and heat-waves likely have even more severe

impacts, but they will also be correspondingly rarer to observe when matched with breeding records and 3 days represents a compromise between a multi-day challenge and enough observation of extreme temperatures to allow us to model fitness effects.”

24. • Related to the above point, I was unclear on why the ECE definition was not carried through to the fitness analyses. The lack of congruence between these analyses makes it difficult to link the results between the sets of analyses and also makes it difficult (as the authors acknowledge) to draw comparisons between species. Was this difference in approach due to the ECE definition in the historical analyses being too strict and thus a lack of ECEs in many of the periods? I think keeping the ECE definition consistent throughout would make for a stronger overall paper, but if there was a clear reason why this was not possible, it should be stated in the Methods.

Based on comments from reviewer 1 above, we have greatly expanded on our ECE definition and clarified our use of temperature thresholds in both the climate and fitness analyses. Given the distinct objectives of these two questions tackled in the paper, it isn't really possible (or desirable) to share thresholds between the two analyses (since only the historical analysis uses a threshold). However, in our initial submission we now see that it was unclear why and when different criteria were used.

Briefly, in the historical analysis, we need a fixed absolute threshold because we wanted to model how the timing of that particular cold or hot threshold changes over time. However, the values we chose previously were not well justified, and we have now set the limits based on the experienced temperatures for the 24 species included in our study (see response above). For the fitness analyses, no threshold is needed since we treat the hottest and coldest period during the attempt as a continuous predictor. In summarizing sensitivity (figure 4) we do make point estimates at 2 SD, which corresponds to about a 1 in 40 event, similar to the ECE definitions in the references suggested by reviewer 1 above. We make this connection explicitly now (see response with added text above). We also include an online appendix that plots the entire GAM response for each species, so that effects at different severities of heat or cold can be examined. We apologize that this additional output wasn't included in the initial submission, but it is now available in the archived data and code repository.

This decision and treatment for the historical analysis vs. the fitness analysis is now explicitly described in the methods.

25. • Given that drawing comparisons between species with particular diets and foraging modes proves important, particularly for points made in the Discussion, it would be helpful to see these distinctions made in the Methods and carried throughout, for example by grouping species in the figures or something along those lines.

Thanks for this suggestion. We are hesitant to put too much emphasis on foraging modes and diet because, as we state in the discussion, we really don't have enough species or enough taxonomic diversity to fully analyze the role of foraging strategy in sensitivity. However, we have added some additional details to the methods and re-worked the main figure (figure 4) showing sensitivity in different stages. We now group species in that figure by rough foraging style when provisioning nestlings (aerial capture, sally foraging, gleaning, and seeds). We also explain those categories in the methods so that the information is presented before we arrive at the discussion. Here is the methods text explaining the addition:

“At this point, we also roughly categorized the included species into four foraging modes based on the primary way that food is obtained during the nestling provisioning period using the species accounts in Birds of the World Online. These included aerial capture (both fly-catching and aerial insectivores), sally foraging (flying to ground from a perch after visually detecting arthropod movement), gleaning (a broad category of strategies involving active search for arthropods), and seed eating. We did not have enough total species or phylogenetic variation in foraging mode to formally analyze diet, but we discuss the possible role that these foraging styles may have had on susceptibility to cold-snaps and heat-waves.”

26. • Line 116 – why was 300 breeding records used as the cut-off and how many species did this exclude?

The exact choice of a cutoff sample size is admittedly somewhat arbitrary. However, the models in our main analysis include spatial effects, year, and data of season along with the coldest and hottest experienced temperature measures. Moreover, we are interested in the effects of relatively rare events. Figure 4 presents fitness for 2 SD heat waves or cold snaps, by definition these should only happen in about 1 out of 40 nesting attempts, so even with a sample of 300 we expect only ~7-8 individual nests below that temperature. Thus, 300 seemed

a bare minimum to have any confidence in the patterns and of course the models run for species with many more nests (e.g., bluebirds, purple martins, and tree swallows) should have much more ability to detect patterns precisely.

Nestwatch allows users to input observations of any species nest. So there are many species that are represented in the larger database by just a handful of nests or even a single nest. We were initially given data by Nestwatch that already had rarely observed species removed so it doesn't really seem accurate to report the number of species excluded by our criteria. We have added this text to the methods:

"We also removed species that had fewer than 300 records. This minimum sample size was somewhat arbitrary, but given the complexity of models that we planned to fit, the desire for spatial and temporal sampling, and the need to model ECEs which, by definition, only occur in a small percentage of nesting attempts, we chose a 300 nest minimum to ensure that we could reliably fit models for each species included. For a species with exactly 300 nests, we would only expect to observe an average of 7.5 nestling attempts that experienced a 2 SD cold-snap or heat-wave."

27. • Line 365 – your analysis currently doesn't account for the grouping of records by weather station and thus the potential similarity in these records as a result of shared conditions. This currently won't be well accounted for by the modelled spatial autocorrelation, so should be added as an additional random effect.

We think that the language in our initial submission was not entirely clear about this analysis. As noted, our nesting data was recorded at individual nests and the weather data was merged based on nest location, meaning that nests located in close proximity would have the same temperature estimates for a given day (as the reviewer notes). Because of this, we chose to estimate fitness grouped by the nearest weather station as we now describe in the Methods section "Impact of cold-snaps and heat-waves on reproductive success" in the lines:

"...weather measurements were indicative of fine-scale regional conditions rather than those measured at specific nest locations, we calculated the response to weather conditions by averaging the number of chicks fledged grouped by hatch date and nearest weather station."

Grouping by weather station in this way is conservative compared to adding a random effect, but it produced for better specified models with GAM checks and also allowed us to model the hatching and fledging from the citizen science data with the exact latitude and longitude of the weather station as spatial covariates to account for spatial autocorrelation (after the grouping at weather station occurred). We view this as a more accurate representation of our data structure since we lacked exact temperatures at nest sites.

28. • There appear to be a few LaTeX-associated issues throughout (e.g., Figure not referred to correctly on lines 217 and 219, zenodo link missing on line 205)

We apologize for the confusion here. We have fixed the rendering issue referring to figures and have added the Zenodo link.

29. • Line 294 – I don't think your results demonstrate that exposure to temperature variability could increase with climate change given that you don't look at the trends in variability, but rather at trends in the timing of ECEs. Therefore, I think you should tone down this point and ensure your conclusions are supported by specific results from your analyses.

Thank you for pointing this out. We think our meaning was just not clear in this sentence. We agree that the current study doesn't examine trends in variability or severity of ECEs. However, what we do show is that springs are warmer on average, but that the timing of cold and hot periods has shifted relatively little. Thus, if birds breed earlier to match average temperatures (not shown in our study but known from many prior studies) then they will necessarily be exposed to a different regime of variation in cold and hot periods. Of course the details here will depend on the species and we agree it is worth toning down the phrasing. We have re-written this sentence in a way that we believe is supported by the specific results of this study:

“Taken together, our results demonstrate that the likelihood of encountering cold-snaps or heat-waves during breeding might differ as a consequence of climate change and that exposure to these events often results in reduced reproductive success for many common and widespread species.”

Comments from Reviewer 3:

30. The manuscript by Taff and Shipley focuses on examining the potential

impacts of cold snaps and heat waves on the North American breeding bird population. The authors have employed a combination of datasets, including local nesting and hatching success data from citizen science initiatives, as well as historical weather records from meteorological stations. This approach allows them to investigate the effects of climatic extremes on population fitness over the past few decades.

The findings of the study indicate that temperature extremes have a detrimental impact on the fitness of approximately half to two-thirds of the species studied. Additionally, for widely distributed species, the authors demonstrate that the effects of these extremes vary based on latitude. By analyzing the occurrence of extreme weather events during the breeding season, the authors reveal that while average spring temperatures have consistently increased throughout North America over time, the patterns of cold snaps and heat waves exhibit more complex dynamics.

In conclusion, the authors emphasize the necessity of acquiring detailed knowledge regarding species' sensitivity to extreme weather events to accurately predict population responses to ongoing climate change.

The manuscript presents valuable and scarce information regarding the link between individual responses to climatic events and population sensitivity. The authors' methodological approach is innovative, straightforward, and has the potential to establish a new standard in the field. The manuscript is well-written and comprehensive. I firmly believe that its publication in *Nature Communications* will garner the appropriate attention and receive a positive response from the research community.

There are a few suggestions that could enhance the manuscript further.

Thank you for these helpful comments. We have responded to each point below.

31. Authors state that all the code will be available alongside with the manuscript, but at the moment is it not, so I cannot see how well it is developed and how easy it would be to re-use it.

We apologize for the oversight. We had meant to include the code with the initial submission but overlooked adding it in the final preparation. It is included now as a Zenodo link (permanent DOI) and is also accessible as a GitHub repository.

32. The methodological details, specifically model equations, are not presented in the text and are sometimes hard to follow.

We have added a schematic of the workflow (see below). We have also included an online appendix that has full model details for every model described in the paper and also the complete set of code with the full model specifications for each model included in the paper. We apologize that these were not included with the initial submission. We had intended to make them available and this was an oversight on our part.

33. I feel that spatially explicit connection between individual and population level could have been developed better. For example authors could map average fitness in species' ranges in the past and now to show where the effects seem to be stronger. They could also compare these maps between the species. Authors could also have looked into spatially explicit predictions of the occurrence of the extreme events and use these as to predict future population fitness. These changes are not necessary, but would nicely conclude the manuscript and would connect the parts with the fitness and spatial maps of extreme events.

We agree this is a fascinating idea! Unfortunately the availability of nest records for the databases we used mostly only goes back to 1995, so we can't really make a historical comparison between a reference period (say 1950-1980 like we used for temperature) vs. a contemporary period. It might be possible to make projections like this using the GAM outputs (now more details included in the online appendix), but only with a lot of assumptions. For example, in order to project back or forward in time, we would need to assume constant sensitivity to cold-snaps and heat-waves (possibly a reasonable assumption), but would also need to project how much each species range and breeding dates have changed or would change under future scenarios in order to determine the temperatures they would actually encounter. In short, we think this is a really fascinating question, but beyond the scope of this paper and perhaps would be better tackled with a few long studied populations with a deeper time series of comparable breeding data from particular locations rather than using the community science databases that we used here.

34. I think that presentation of fitness effects of extremes followed by their spatial distribution would improve the manuscript flow.

We appreciate this comment, but we prefer to keep the current order as starting with the historical analysis establishes the importance of ECEs in the context of climate change, before then demonstrating fitness effects for common birds. Ultimately, this is probably a matter of personal preference as the results could be presented in either order effectively.

35. L144 - Are you utilizing Natural Earth Data for any analysis? If not, it may not be necessary to mention it here.

While it isn't used for any statistical analyses, the basemap (land and ocean outlines) is accessed through the R natural earth package. Since those layers are used in our figures we think it is still appropriate to cite the package.

36. L147 - Consider choosing equal area hexagons instead of 2-degree intervals. The former would facilitate easier comparisons and extrapolation.

We have changed the analysis to use equal area hexagons rather than 2-degree hexagons. This doesn't result in any qualitative changes to results and in fact most of the main analyses don't rely on data summarized at the hexagon level so they are unchanged. All the figures are now redrawn with equal area hexagons.

37. L225-232 - It would greatly enhance the manuscript to include a schematic of the analysis or, at the very least, provide some equations used.

We have included a schematic of the entire workflow as a supplemental figure that is now referenced in the data section of the methods. The schematic also helps to orient the data and code repository structure.

38. L234 - It is worth noting that the first and last events may not always follow a normal distribution. Therefore, it is essential to perform checks on the error distribution before utilizing gams with Gaussian errors.

We completely agree, and for this particular model, it isn't a first and last event timing in the response, but rather the average (within each grid cell) of the first heat-wave and last cold-snap anomaly over a period of 25 years. From our understanding this should minimize the influence of non-normal first and last events that would be better handled by a different model or distribution (i.e. cox proportional hazards model or similar form). For all of the models, we performed diagnostic tests on the models to ensure they met our initial assumptions of the error distribution using the tools provided within the R package 'mgcViz'. We have provided the residual plots for both the cold anomaly and heat anomaly models below on the left and right, respectively. Although there are a few outliers, the mean is centered around zero and shows little evidence of violating our assumption of normality.

39. L249 - Could you clarify the error distribution used in this particular case?

These were fit with a Gaussian error distribution and we have modified the sentence following this section to make it clear that it applies to both nestling and provisioning models:

“We implemented a Gaussian model for both the incubation and nestling provisioning models...”

40. L258-259 - While Poisson models are susceptible to over-dispersion, it would be helpful to know if you encountered this issue. Additionally, incorporating individual random effects can address this problem. It is also more common to switch to a negative-binomial distribution rather than a Gaussian one.

We have noted in the methods section that we initially encountered overdispersion when modeling as a Poisson process. However, as also noted above - due to the spatial mismatch between the nest observation data and the weather stations, we decided to model the data as the mean fitness relative to each weather station, grouped by weather station and hatch date, to more accurately represent the structure of our dataset. By doing this, the data on fitness measures is no longer count based, but rather a continuous response. We have checked carefully in the revision to ensure that we provide details on distributions and checking for model assumptions. We also now provide the complete set of code in an archived repository and complete model output for each described model in an online appendix with the code repository.

41. L307 - In Figure 1, consider using different color schemes for the three maps. Alternatively, if using a single palette, color them based on their effects on birds, such as assigning the same positive color to early cold snaps and later heat waves.

We have changed these figures to have a different diverging color scheme in each panel as suggested.

42. L307 - Furthermore, it would be interesting to include the duration of the average breeding season (between the last cold snap and the first heat wave) in Figure 1. This could provide insights into whether the duration becomes critically short in certain locations.

We agree that it would be very interesting to look at the change in breeding season duration spatially, but think that this is beyond the scope of the current study. We could make plots for the duration between first cold snap and last heat wave and the anomaly in length of that duration from 1995-2020 compared to 1950-1980. However, we would need to choose temperature values to use (which is tricky as described in many of the comments above!) and this could quickly become a large and unwieldy multi-panel plot. It seems to us that this particular question of change in breeding duration would be better conducted at a per species level based on species specific sensitivity to ECEs and the range and timing of their breeding. While important, that approach would perhaps be best tackled in a different paper focused on the length of the breeding season specifically rather than the consequences of ECE exposure.

43. L317 - There appears to be an issue with the figure at this point (no figure number and no figure caption).

We apologize for the error here. This was an issue with automatic figure numbering when the manuscript was rendered that also caused the caption not to print. It has been fixed.

REVIEWER COMMENTS

Reviewer #1 (Remarks to the Author):

With pleasure I have done a second review of the manuscript titled "Inconsistent shifts in warming and temperature variability negatively impact avian fitness". The authors have done an impressive job in considering and addressing all of my concerns during my initial review. A majority of this is reflected in the definitions, terminology, and new analyses that are presented in this second form. Therefore, I am pleased with the quality of the content in this version of the manuscript. Below I just have a few minor edits to consider.

Line 142: Consider changing to "In cases where only a laying date was provided, we inferred approximate hatching dates for successful nests by...". This is probably implied but just make it explicit.

Figure 2: Please specify the units on these threshold values in panels B and C. As presented, that information is lacking.

Line 399: You don't have an open parenthesis.

Reviewer #2 (Remarks to the Author):

I want to thank the authors for their thorough responses to the suggestions made by myself and the other reviewers. The responses have served to clear multiple things up, especially when it comes to the methods, and I think the manuscript is undoubtedly stronger for the changes made. I just have one remaining question that relates to one of my previous queries:

I previously had some questions regarding the grouping of data at the same weather stations and potential pseudoreplication. Your response and associated changes to the manuscript text clarified this somewhat. However, unless I've again misunderstood, it still seems that there may be a problem with pseudoreplication. Given that your data are grouped by hatch date and weather station to produce the data used for analysis, is it not the case that data points grouped by weather station but with different hatch dates are being treated as independent data points when they're inherently not as they share the same weather station? Thus, I think that my comment about the need for a random effect for weather station may still stand.

Comments from Reviewer #1:

1. With pleasure I have done a second review of the manuscript titled "Inconsistent shifts in warming and temperature variability negatively impact avian fitness". The authors have done an impressive job in considering and addressing all of my concerns during my initial review. A majority of this is reflected in the definitions, terminology, and new analyses that are presented in this second form. Therefore, I am pleased with the quality of the content in this version of the manuscript. Below I just have a few minor edits to consider.

Thank you again for the helpful comments on our manuscript.

2. Line 142: Consider changing to "In cases where only a laying date was provided, we inferred approximate hatching dates for successful nests by...". This is probably implied but just make it explicit.

We have made this change exactly as suggested.

3. Figure 2: Please specify the units on these threshold values in panels B and C. As presented, that information is lacking.

We have made this change as suggested.

4. Line 399: You don't have an open parenthesis.

We have fixed this mistake.

Comments from Reviewer #2:

5. I want to thank the authors for their thorough responses to the suggestions made by myself and the other reviewers. The responses have served to clear multiple things up, especially when it comes to the methods, and I think the manuscript is undoubtedly stronger for the changes made. I just have one remaining question that relates to one of my previous queries:

Thank you for reviewing the paper again. We agree that the revision is stronger as a result of the thoughtful comments on our submission.

6. I previously had some questions regarding the grouping of data at the same weather stations and potential pseudoreplication. Your response and associated changes to the manuscript text clarified this somewhat. However, unless I've again misunderstood, it still seems that there may be a problem with pseudoreplication. Given that your data are grouped by hatch date and weather station to produce the data used for analysis, is it not the case that data points grouped by weather station but with different hatch dates are being treated as independent data points when they're inherently not as they share the same weather station? Thus, I think that my comment about the need for a random effect for weather station may still stand.

Thank you for prompting us to think carefully about the model structure we employed in light of this comment. We appreciate and agree with the suggestion that there will almost certainly be local autocorrelation of nest outcomes from observations given the way that nests are distributed in space. However, we respectfully disagree with the argument that including weather station as a random effect is an appropriate or necessary solution to this issue for several reasons that we describe below.

First, we do include spatial smooths for latitude and longitude that should capture spatial autocorrelation. Our goal is to estimate overall species level sensitivity to cold and heat while accounting for spatial patterns, and we are confident that this is what our model structure accomplishes. One of the key challenges in modeling this data is accounting for spatial autocorrelation that is likely present, as our dataset often extends across the entire species range. Inclement weather events are inherently spatial, and therefore regional phenomena that affect fitness are likely to affect nearby sites more than more distant ones. Adding a localized random effect (e.g., station id) is, to our thinking, an attempt to account for the same issue of spatial non-independence, but because it would be a dimensionless factor it is unclear how to interpret these results knowing station id is inherently spatial while also including latitude and longitude information in the same model. Thus, we think the addition of this random effect would add complexity and complicate interpretation unnecessarily.

We would also point out here that we do not think it is appropriate to describe this as a choice that is necessary to account for pseudoreplication, since individual nesting attempts are only observed and included once in the dataset (in fact we are already quite conservative in averaging observations of the same species that hatched on the same day with a shared weather station). In no case was there more than one observation at a site per day and in no case were the same birds counted more than once, as our data is based on individual nest reports. Rather, the issue is to appropriately deal with spatial non-independence in order to allow us to make the best overall range-wide estimates, which we accomplished by latitude and longitude spatial smooths.

Second, even if we want to construct a model that allows for random intercepts for different cluster levels, it isn't clear to us that station id is a good candidate to capture this heterogeneity. Presumably, the non-independence of nearby nesting attempts that the reviewer notes would result from local unmeasured variables and not from the weather station used per se, since there is no way that we can think of in which the weather station identity would causally influence the response variable (number of fledged offspring). Moreover, if there are local unmeasured factors that influence fitness, it isn't clear to us that weather station would serve as a good target for a stable factor across years, because we cannot think of any plausible mechanism by which weather station identity would have a stable causal influence (across years) on fledging success. It might be more appropriate to include a local random effect nested within year (e.g., weather station by year factor), however this specification would exacerbate the model fitting issue we describe next. By the same argument, any number of other potential grouping factors at a local scale could be included (e.g., subnational region or habitat type). Generally speaking, inclusion of a random effect should be justified by experimental design

involved in data collection (which does not apply here) or by a suspected causal link between the different levels and the measured outcome (e.g., school attendance as a random effect for test scores); we do not think station id meets either of those criteria in this case.

Third, one of our key concerns with using station id as a random effect is the ratio of nest observations per weather station, especially for species with fewer data points. Because of this issue, the data for most species are not well suited to including this random effect. Despite the large overall sample size in our dataset, we accessed weather data from so many weather stations that individual stations are often matched with only one or a few nests. This is especially true for species with modest sample sizes, but even for the species with large sample sizes a significant portion of weather stations are only matched to one or two separate nests. Although it is acceptable to have single observations per random effect level (Maas and Hox, 2005), it does significantly limit the statistical power to estimate those random effects and their meaningfulness. We include a table illustrating this issue when considering weather station id alone or nested within year (Table i). We show that in many cases more than half of the levels (station id) have only one or two nest observations and that the percentage of levels with few observations is even more extreme for station nested in year. The large number of levels to estimate coupled with few observations per level creates model convergence issues in some cases, although for models that do behave the results are generally qualitatively similar to our original specification.

Table i. Summary of observations by station id and station id nested within year. Numbers shown here are for all nests, but the pattern is similar when considering only nests that hatched.

Species	Station id				Station id nested in year		
	Total nests	Levels	Observation to level ratio	Percent levels with <3 observations	Levels	Observation to level ratio	Percent levels with <3 observations
American robin	1672	438	3.8	58%	1028	1.6	88%
Barn swallow	624	176	3.5	69%	303	2.1	82%
Bewick's wren	1108	83	13.3	51%	279	4	60%
Black crested titmouse	379	15	25.3	49%	80	4.7	79%
Black-capped chickadee	2492	377	6.6	49%	1239	2	79%
Brown-headed nuthatch	338	43	7.9	47%	130	2.6	71%
Carolina chickadee	3800	293	13	39%	1282	3	68%
Chestnut-backed chickadee	621	28	22.2	61%	108	5.8	45%
Eastern bluebird	53422	844	63.3	20%	5383	9.9	38%
Eastern Phoebe	867	245	3.5	62%	547	1.6	90%
House finch	543	240	2.3	74%	414	1.3	93%
House wren	12940	514	25.2	33%	2474	5.2	49%
Mountain bluebird	4138	96	43.1	39%	421	9.8	34%
Mountain chickadee	388	38	10.2	42%	152	2.6	66%
Mourning dove	459	216	2.1	78%	327	1.4	92%
Northern cardinal	441	229	1.9	79%	376	1.2	97%
Oak titmouse	562	26	21.6	31%	153	3.7	57%
Prothonotary warbler	660	41	16.1	51%	115	5.7	56%
Purple martin	30456	364	83.7	3%	2255	13.5	9%
Tree swallow	29065	653	44.5	23%	3709	7.8	38%
Tufted titmouse	1118	214	5.2	60%	589	1.9	81%
Violet green swallow	1401	85	16.5	39%	360	3.9	60%
Western bluebird	7698	130	59.2	28%	666	11.6	36%
White-breasted nuthatch	337	78	4.3	65%	214	1.6	87%

Given these arguments, we prefer to retain the models as originally presented without station id as a random effect in the manuscript. We want to make it clear, however, that this decision is based on what we take to be design considerations rather than a difference in results. Apart from some convergence issues and a few differences of significance attributable to larger confidence intervals, the results that we presented are qualitatively similar when including a random effect. For example, one alternative way to encode random effects similar to those suggested by the reviewer might be to use the hexagon grid identity from our summary maps as a random effect. This specification accomplishes a similar goal of grouping spatially clustered nests but avoids some of our arguments for points 2 and 3 above, although we still think that our point 1 argues against it and this is still not fit with grid nested in year to avoid estimation issues. In any case, refitting our main set of fitness models using grid id as a random effect yields nearly identical point estimates and qualitatively similar significance estimates for the majority of species (Figure i).

Figure i. Main analysis species relative fitness models from Figure 4 of the original submission refit with grid id as a random effect.

Ultimately, we think that there is room for reasonable debate about analysis choices for a complex dataset like the one we use here. We have tried to specify and justify our choices transparently. We are happy to engage in more dialogue if the editor or reviewer would still like to see a version of this random effect added in the final version of analysis for the paper, but our opinion is that the current analysis has a stronger justification.